# VOCE: Variational Optimization with Conservative Estimation for Offline Safe Reinforcement Learning

**Jiayi Guan**
Tongji University
*guanjiayi@tongji.edu.cn*

**Guang Chen**[*]
Tongji University
*guangchen@tongji.edu.cn*

**Jiaming Ji**
Peking University
*jiamg.ji@gmail.com*

**Long Yang**
Peking University
*yanglong001@pku.edu.cn*

**Ao Zhou**
Tongji University
*2211107@tongji.edu.cn*

**Zhijun Li**
Tongji University
*zjli@ieee.org*

**Changjun Jiang**
Tongji University
*cjjiang@tongji.edu.cn*

## Abstract

Offline safe reinforcement learning (RL) algorithms promise to learn policies that satisfy safety constraints directly in offline datasets without interacting with the environment. This arrangement is particularly important in scenarios with high sampling costs and potential dangers, such as autonomous driving and robotics. However, the influence of safety constraints and out-of-distribution (OOD) actions have made it challenging for previous methods to achieve high reward returns while ensuring safety. In this work, we propose a **V**ariational **O**ptimization with **C**onservative **E**stimation algorithm (VOCE) to solve the problem of optimizing safety policies in the offline dataset. Concretely, we reframe the problem of offline safe RL using probabilistic inference, which introduces variational distributions to make the optimization of policies more flexible. Subsequently, we utilize pessimistic estimation methods to estimate the Q-value of cost and reward, which mitigates the extrapolation errors induced by OOD actions. Finally, extensive experiments demonstrate that the VOCE algorithm achieves competitive performance across multiple experimental tasks, particularly outperforming state-of-the-art algorithms in terms of safety. Our code is available at github.VOCE.

## 1 Introduction

Reinforcement learning (RL) algorithms have made remarkable achievements in various fields such as robot control [1–3] and strategy games [4–6]. However, limited by the online and trial-and-error nature, standard RL is challenging to apply to dangerous and expensive scenarios [7–9]. Offline safe RL is a promising and potential approach to tackle the above problem, which learns policy satisfying safety constraints from pre-collected offline datasets without interacting with the environment [10, 11].

Since offline safe RL methods aim to learn policies that satisfy safety constraints from offline datasets, which requires the agent to not only comply with safety constraints but also consider the influence of out-of-distribution (OOD) actions [12–15]. This makes it difficult for the algorithm to learn policies that achieve high reward returns while satisfying safety constraints [16]. There are two main

---

[*]Corresponding author

37th Conference on Neural Information Processing Systems (NeurIPS 2023).

approaches to deal with the aforementioned challenges: one is based on linear programming [17, 18], and the other is based on exploration and evaluation [19–21]. Linear programming provides a way to research offline safe RL problems, but it heavily depends on the F-divergence or KL-divergence to constrain the distance between the optimal distribution and the sample distribution [17]. Therefore, it is difficult for linear programming to eliminate the extrapolation errors caused by OOD actions [10, 11]. The exploration evaluation approaches are to introduce conservative Q-value estimation under the actor-critic framework. Although exploration evaluation approaches avoid the overestimation issue through lower bound Q-values, they prematurely terminate trajectories that exceed the constraints during policy updates [16]. The exploration evaluation method with strict constraints on the sample space makes it challenging to search for reward-maximizing policies. In conclusion, existing methods face challenges in learning policies from offline datasets that maximize reward while satisfying safety constraints, particularly when considering the impact of extrapolation errors.

In this work, to solve the constrained policy optimization problem and eliminate the extrapolation errors caused by OOD actions, we propose a variational optimization with conservative estimates for the offline safe RL algorithm. Concretely, we reframe the objective of offline safe RL using probabilistic inference, enhancing the flexibility of policy optimization by replacing parameterized policies with variational distributions. Furthermore, to avoid extrapolation errors caused by OOD actions, we derive upper and lower bounds for Q-values and estimate the Q-values of costs and rewards based on these bounds. Finally, we adopt a supervised learning approach to train a parameterized policy network according to the variational distribution of policies. The main contributions of this work are listed as follows:

- We utilize probabilistic inference to address the problem of offline safe RL, which introduces non-parametric variational distributions to replace parameterized policies, providing increased flexibility for optimizing safe policies in offline datasets.

- We derived upper and lower bounds for Q-value estimation using the pessimistic estimation approach. Subsequently, we utilized these bounds to estimate the Q-value of costs and rewards, respectively, to avoid extrapolation errors caused by OOD actions.

- We carry out extensive comparative experiments, and the results indicate that the VOCE algorithm outperforms state-of-the-art algorithms, especially in terms of safety.

## 2 Preliminaries

Constrained Markov decision processes (CMDP) provide a theoretical framework to solve safe RL problems [22], where the agent is cost-bounded by safety constraints. A CMDP is defined as a tuple $(\mathcal{S}, \mathcal{A}, C, P, r, \rho_0, \gamma)$, where $\mathcal{S} \in \mathbb{R}^n$ is the state space, $\mathcal{A} \in \mathbb{R}^m$ is the action space, $P : \mathcal{S} \times \mathcal{A} \times \mathcal{S} \to [0, 1]$ is the transition kernel, which specifies the transition probability $p(s_{t+1}|s_t, a_t)$ from state $s_t$ to state $s_{t+1}$ under the action $a_t$, $r : \mathcal{S} \times \mathcal{A} \to \mathbb{R}$ represents the reward function, $C$ is the set of costs $\{c_i : \mathcal{S} \times \mathcal{A} \to \mathbb{R}_+, i = 1, 2, \cdots, m\}$ for violating $m$ constraints, $\gamma \in (0, 1]$ is the discount factor, and $\rho_0 : \mathcal{S} \to [0, 1]$ is the distribution of initial states. A policy $\pi$ maps a probability distribution from state $s_t$ to action $a_t$. We utilize $\pi_\theta$ to denote the parameterization of the policy with parameter $\theta$. In safe RL, the goal is to maximize the cumulative reward while satisfying safe constraints:

$$\pi^* = \arg\max_\pi \mathbb{E}_{\tau \sim \pi} \left[ \sum_{t=0}^\infty \gamma^t r(s_t, a_t) \right], \quad \text{s.t.} \quad \mathbb{E}_{\tau \sim \pi} \left[ \sum_{t=0}^\infty \gamma^t c_i(s_t, a_t) \right] \le \bar{c}_i, \tag{1}$$

where the $\tau = \{s_0, a_0, \cdots\} \sim \pi$ denotes the trajectory. $\bar{c}_i$ is the $i$-th safe constraint limit.

In the offline RL setting, we learn a policy according to the offline dataset $D$ collected by one or more data-collection policies, and without online interaction with the environment ($D = \{(s_t, a_t, r_t, c_t)_i\}_{i=1}^n$). Although this way of not interacting with the environment brings a lot of advantages, the offline dataset can not cover all action-state transitions, and the policy evaluation step actually utilizes the Bellman equation of a single sample [23]. This makes the estimated Q-value vulnerable to OOD actions, which in turn severely affects the performance of the algorithm [24, 25]. As shown in Fig. 1, under the offline setting, off-policy safe RL methods neither learn a high-reward policy nor guarantee safety. Additionally, although offline RL methods obtain high rewards under safe expert data, it is difficult to guarantee safety because it directly ignores the cost. We can conclude that both offline RL and safe RL face challenges in learning policies that satisfy safety constraints

from offline datasets. Therefore, it is important to design an algorithm that learns high-reward and satisfies safety constraints, which is the focus of this paper.

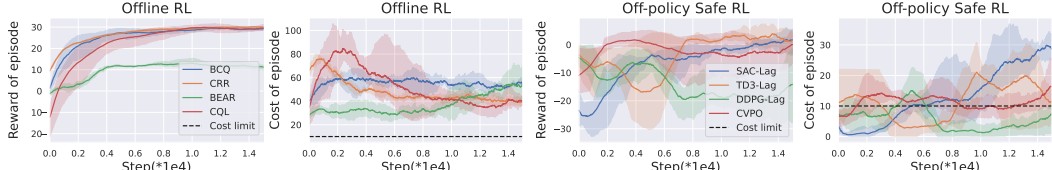

Figure 1: Reward and cost curves of safe RL and off-policy RL under high-reward and safe datasets in *Point-Button* task [26]. The solid line is the mean of the performance, and the shaded line represents the variance of the performance. All performance curves employ more than 3 random seeds.

## 3  Method

In this section, we present the details of Variationally Optimized with Conservatively Estimated for offline-safe RL algorithm (VOCE), which is the main contribution of this paper. We first reformulate the objective of offline-safe RL with probabilistic inference. Based on this, we derived upper and lower bounds for Q-value estimation, and adopted a pessimistic and conservative estimation approach to estimate the Q-value of costs and rewards respectively to eliminate the extrapolation error of OOD actions. Finally, we employ supervised learning to transform the variational distribution into a parameterized policy.

### 3.1  Offline-safe RL as Inference

From the probabilistic inference perspective, the offline safe RL can be viewed as the problem of observing safe actions in states with high rewards. As the probabilistic inference model and notation in [26, 27], we introduce an optimal variable $\mathcal{O}$ to represent the event of maximizing rewards. Assuming the likelihood of a given trajectory being optimal is proportional to the cumulative discounted reward, the infinite discounted reward formula is expressed as $\mathcal{P}(\mathcal{O}=1|\tau) \propto \sum_t \gamma^t r(s_t, a_t)$. Since $\mathcal{P}(\mathcal{O}=1|\tau) \geq 0$, we further confirm that the likelihood of a given trajectory being optimal is proportional to the exponential of the accumulative discounted reward, then we rewrite the infinite discounted reward formula as $\mathcal{P}(\mathcal{O}=1|\tau) \propto \exp\left(\sum_t \gamma^t r_t/\alpha\right)$. The $\alpha$ is a temperature parameter, and $r_t$ is short for $r(s_t, a_t)$. Let $P_\pi(\tau)$ be the probability of the trajectory under the policy $\pi$, then the log-likelihood of optimality under policy $\pi$ can be written as $\log \mathcal{P}_\pi(\mathcal{O}=1) = \log \sum_\tau \mathcal{P}(\mathcal{O}=1|\tau)P_\pi(\tau)$. Furthermore, according to importance sampling and Jensen's inequality, we obtain a lower bound of the log-likelihood of optimality under policy $\pi$.

$$\log \mathcal{P}_\pi(\mathcal{O}=1) = \log \mathbb{E}_{\tau \sim \mathcal{Q}}\left[\frac{\mathcal{P}(\mathcal{O}=1|\tau)P_\pi(\tau)}{\mathcal{Q}(\tau)}\right] \geq \mathbb{E}_{\tau \sim \mathcal{Q}} \log \frac{\mathcal{P}(\mathcal{O}=1|\tau)P_\pi(\tau)}{\mathcal{Q}(\tau)}$$
$$\propto \mathbb{E}_{\tau \sim \mathcal{Q}}\Big[\sum_{t=0}^\infty \gamma^t r_t\Big] - \alpha D_{KL}(\mathcal{Q}(\tau)||P_\pi(\tau)), \tag{2}$$

where the $\mathcal{Q}(\tau)$ is the auxiliary trajectory distribution. Since offline safe RL not only needs to consider maximizing cumulative rewards but also guaranteeing safety, we restrict $\mathcal{Q}(\tau)$ to the constrained feasible distribution space. According to the constraint threshold $\bar{c}_i$, we define the feasible distribution space as:

$$\Omega^{\bar{c}_i} \triangleq \{q(a_t|s_t) : \mathbb{E}_{\tau \sim q}[\sum_{t=0}^\infty \gamma^t c_i(s_t, a_t)] \leq \bar{c}_i\}, \tag{3}$$

where each $q(a_t|s_t) \in \Omega^{\bar{c}_i}$ indicates that the action distribution of the sate-condition satisfies the safety constraint. By factorizing the trajectory distribution [27], we express the trajectory distribution as:

$$\mathcal{Q}(\tau) = \rho(s_0)\prod_{t \geq 0} p(s_{t+1}|s_t, a_t)q(a_t|s_t), q(a_t|s_t) \in \Omega^{\bar{c}_i}, \tag{4}$$

$$P_{\pi_\theta}(\tau) = \rho(s_0)\prod_{t \geq 0} p(s_{t+1}|s_t, a_t)\pi_\theta(a_t|s_t), \tag{5}$$

where $\rho(s_0)$ is the distribution of the initial state $s_0$. Substituting Eq. (4) and (5) into Eq. (2) to eliminate the transitions $p(s_{t+1}|s_t, a_t)$, and combining the feasible distribution space of Eq. (3), we obtain the object of offline safe RL problem as shown in Proposition 3.1.

**Proposition 3.1.** *The objective of the offline safe RL problem can be defined through probabilistic inference as follows:*

$$\mathcal{L}(q, \pi) = \mathbb{E}_{\tau \sim q}\Big[ \sum_{t=0}^{\infty}(\gamma^t r_t) - \alpha D_{KL}(q(\cdot|s_t)||\pi_\theta(\cdot|s_t))\Big], \ q(a_t|s_t) \in \Omega^{\bar{c}_i}. \tag{6}$$

Proposition 3.1 provides a new objective for the problem of offline safe RL. The aforementioned probabilistic inference method has many optimizations over the previous dual methods in solving the problem of offline safe RL. This approach provides flexibility for optimizing the optimal policy by introducing a variational distribution $q(a_t|s_t)$ to connect the strong correlation between dual variable optimization and policy improvement. Furthermore, the approach decouples the optimal policy optimization problem into independent processes of optimal variational distribution optimization and parameterized policy update. We will introduce these two update processes in detail later.

### 3.2 Variational Optimization with Conservative Estimates for Offline-safe RL

The previous standard RL and safe RL also employed the above-mentioned idea of probabilistic inference to solve the policy optimization problem [26, 28]. However, due to the influence of OOD actions in the offline setting, previous RL algorithms based on probabilistic inference are difficult to address the tasks of offline-safe RL. Therefore, we adopt a pessimistic conservative estimation method to improve the offline-safe RL algorithm based on probabilistic inference and obtain a **V**ariational **O**ptimization with **C**onservative **E**stimation algorithm (VOCE). We divide the VOCE into two steps: conservatively estimated variational optimization and parameterized policy update.

#### 3.2.1 Conservatively Estimated Variational Optimization

The goal of the conservatively estimated variational optimization is to optimize the optimal variational distribution $q(a_t|s_t)$ with a high reward while satisfying the safety constraints. In this step, we perform the maximization of the variational distribution $q(a_t|s_t)$ by fixing the policy $\pi_\theta(a_t|s_t)$. According to proposition 3.1, we write the objective of solving the optimal variational distribution as:

$$\mathcal{L}(q) = \mathbb{E}_{\rho(s_0)}\Big[\mathbb{E}_{q(a_t|s_t)}Q^r(s_t, a_t) - \alpha D_{KL}(q(\cdot|s_t)||\pi_\theta(\cdot|s_t))\Big], \quad \text{s.t.} \mathbb{E}_{\rho(s_0)}\Big[\mathbb{E}_{q(a_t|s_t)}Q^{c_i}(s_t, a_t)\Big] \leq \bar{c}_i. \tag{7}$$

The optimization objective of the above variational distribution can be regarded as a KL-regularized constrained optimization problem. However, since the expected reward term $\mathbb{E}_{q(a_t|s_t)}Q^r(s_t, a_t)$ may be on different scales depending on the environment, it is difficult to set an appropriate penalty coefficient $\alpha$ for different environments. Therefore, we introduce hard constraints to replace the regularization term of the KL-divergence and rewrite the optimization objective of Eq. (7) as:

$$\max_q \mathbb{E}_{\rho(s_0)}\Big[ \sum_{a_t} q(a_t|s_t)Q^r(s_t, a_t)\Big], \quad \text{s.t.} \ \mathbb{E}_{\rho(s_0)}\Big[ \sum_{a_t} q(a_t|s_t)Q^{c_i}(s_t, a_t)\Big] \leq \bar{c}_i,$$

$$\mathbb{E}_{\rho(s_0)}\Big[D_{KL}(q(a_t|s_t)||\pi_\theta(a_t|s_t))\Big] \leq \epsilon, \quad \sum_{a_t} q(a_t|s_t) = 1, \forall s_t \in \mathcal{D}, \tag{8}$$

where $\epsilon$ is the KL-divergence threshold between the variational distribution $q(a_t|s_t)$ and the parameterized policy $\pi_\theta(a_t|s_t)$. To solve the above-constrained problem (8), we replace the parametric $q(a_t|s_t)$ with the non-parametric to avoid performance degradation caused by approximation errors [29]. Furthermore, we apply the Lagrange multiplier method to transform the above-constrained problem into an unconstrained problem. In the non-parametric form $q(a_t|s_t)$, since the objective function is linear and the constraints are convex, the constrained optimization problem shown in Eq. (8) is a convex optimization problem. Furthermore, through mild assumptions, we can obtain the strong dual form of Eq. (9).

**Assumption 3.2.** *(Slater's condition). There exists a variational distribution that satisfies the safety constraints $q(a_t|s_t) \in \Omega^{\bar{c}_i}$ within the current policy trust region $D_{KL}(q(a_t|s_t)||\pi_\theta(a_t|s_t)) < \epsilon$.*

**Lemma 3.3.** *Under Assumption 3.2, there exists a strong duality between the primal problem in Eq. (8) and the unconstrained problem in Eq. (9).*

$$\mathcal{L}(q, \lambda, \eta, \delta) = \min_{\lambda \geq 0, \eta \geq 0, \delta} \max_{q \geq 0} \mathbb{E}_{\rho(s_0)} \left[ \sum_{a_t} q(a_t|s_t) Q^r(s_t, a_t) - \lambda^T \left[ \sum_{a_t} q(a_t|s_t) Q^{c_i}(s_t, a_t) - \bar{c}_i \right] \right.$$
$$\left. - \delta \left[ \sum_{a_t} q(a_t|s_t) - 1 \right] - \eta \left[ D_{KL}\left( q(a_t|s_t) || \pi_\theta(a_t|s_t) \right) - \epsilon \right] \right],$$
(9)

where $\lambda, \eta, \delta$ are the Lagrange multipliers of the above multiple constraints. Based on the strong duality of the unconstrained problem in Eq. (9), we further derive a closed-form solution $q^*(a_t|s_t)$ for its internal maximization, as shown in Proposition 3.4. Proofs and discussions are in Appendix B.1.

**Proposition 3.4.** *The closed-form solution of the optimal variational distribution that satisfies the safety constraint in Eq. (9) is given as:*

$$q^*(a_t|s_t) = \pi_\theta(a_t|s_t) \exp\left[ \frac{Q^{rc}(s_t, a_t)}{\eta} \right] \exp\left( -\frac{\eta + \delta}{\eta} \right),$$
(10)

*where $Q^{rc}(s_t, a_t) \triangleq Q^r(s_t, a_t) - \lambda^T Q^{c_i}(s_t, a_t)$.*

By substituting the closed-form solution (10) into Eq. (9) and eliminating similar terms, we obtain the convex minimization problem shown in Proposition 3.5. Subsequently, we solve for the dual variables $\eta$ and $\lambda$ using Eq. (11). Proofs and discussions are in Appendix B.2.

**Proposition 3.5.** *The $\lambda, \eta$ are dual variables and are solved via the convex optimization problem of the following formula.*

$$\mathcal{L}(\lambda, \eta) = \min_{\lambda \geq 0, \eta \geq 0} \mathbb{E}_{\rho(s_0)} \left[ \eta \log \sum_{a_t} \pi_\theta(a_t|s_t) \exp\left( \frac{Q^{rc}(s_t, a_t)}{\eta} \right) + \lambda^T \bar{c}_i + \eta \epsilon \right].$$
(11)

The Proposition 3.4 provides a proposal to solve a non-parametric variational distribution $q(a_t|s_t)$ for the given the Q-value $Q^r(s_t, a_t)$ and $Q^{c_i}(s_t, a_t)$. In addition, we provide the optimality and uniqueness of the above closed-form $q^*(a_t|s_t)$ solution under the premise of strong convexity in the Appendix B.2. Note that it can be seen from Eq. (10) that providing accurate $Q^r(s_t, a_t)$ and $Q^{c_i}(s_t, a_t)$ are the premise and guarantee for accurately computing the non-parametric variational distribution $q(a_t|s_t)$. We can find similar conclusions in the online safe RL algorithm CVPO [26], as described in Proposition 3.4 and 3.5. In the online setting, the empirical Bellman equation is directly applied to iteratively update the Q-value. However, in the offline setting, this approach would lead to clear extrapolation errors in $Q^{c_i}(s_t, a_t)$ and $Q^r(s_t, a_t)$ due to the OOD actions. Furthermore, it is difficult to accurately compute the variational distribution $q(a_t|s_t)$ according to Eq. (10).

To eliminate the extrapolation error caused by the OOD actions during the evaluation of the Q-value, we utilize the pessimistic conservative estimation approach to estimate $Q^r$ and $Q^{c_i}$, respectively. Specifically, to eliminate the impact of extrapolation error on $Q^r$, we need to avoid overestimation of $Q^r$ [24]. Therefore, similar to CQL [30] to learn a conservative lower bound of the Q-function by additionally minimizing the Q-values alongside a standard Bellman error objective, we choose the penalty that minimizes the expected Q-value under special action-state transitions for unknown actions produced by $\pi_\mathcal{M}(a_t|s_t)$. Then, we define the Q-value of the iterative reward function as:

$$\hat{Q}^r_{k+1} \leftarrow \arg\min \frac{1}{2} \mathbb{E}_{s_t, a_t \sim D} \left[ \left[ (Q^r(s_t, a_t)) - \hat{\mathcal{B}}^\pi \hat{Q}^r_k(s_t, a_t) \right]^2 \right] + \kappa \mathbb{E}_{s_t \sim D, a_t \sim \pi_\mathcal{M}(\cdot|s_t)} \left[ Q^r(s_t, a_t) \right],$$
(12)

where the $\hat{\mathcal{B}}^\pi \hat{Q}^r(s_t, a_t) = r(s_t, a_t) + \gamma \hat{Q}^r(s_{t+1}, a_{t+1})$. In Proposition 3.6, we show that $\hat{Q}^r(s_t, a_t)$ converges to a lower bound on $\hat{Q}^r(s_t, a_t)$. However, we can tighten this bound if we are interested in $V^r(s_t)$. We improve the bounds by introducing an additional maximization term under the sample distribution $\hat{\pi}_\beta$. Then the iterative update Eq. (12) of the reward Q-value can be rewritten as:

$$\hat{Q}^r_{k+1} \leftarrow \arg\min \frac{1}{2} \mathbb{E}_{s_t, a_t \sim D} \left[ \left[ (Q^r(s_t, a_t)) - \hat{\mathcal{B}}^\pi \hat{Q}^r_k(s_t, a_t) \right]^2 \right] +$$
$$\kappa \left[ \mathbb{E}_{\substack{s_t \sim D \\ a_t \sim \pi_\mathcal{M}(\cdot|s_t)}} Q^r(s_t, a_t) - \mathbb{E}_{\substack{s_t \sim D \\ a_t \sim \hat{\pi}_\beta(\cdot|s_t)}} Q^r(s_t, a_t) \right],$$
(13)

where $\kappa$ is a tradeoff factor. Note that the Eq. (12) and (13) utilize the empirical Bellman operator $\hat{\mathcal{B}}^\pi$ instead of the actual Bellman Operator $\mathcal{B}^\pi$. Following the related work [23, 31], we employ the

concentration properties to compute the error. For any $\forall s_t, a_t \in D$, with the probability $\geq 1 - \delta$, the sampling error can be written as:

$$|\mathcal{B}^\pi \hat{Q}(s_t, a_t) - \hat{\mathcal{B}}^\pi \hat{Q}(s_t, a_t)| \leq \frac{C_{r,p,\delta} R_{max}}{(1-\gamma)\sqrt{|\mathcal{D}(s_t, a_t)|}}, \tag{14}$$

where $C_{r,p,\delta}$ is a constant depending on the concentration properties of $r(s_t, a_t)$, $p(s_{t+1}|s_t, a_t)$ and the $\delta, \delta \in (0,1)$. The $\frac{1}{\sqrt{|\mathcal{D}(s_t,a_t)|}}$ represents a vector of size $|\mathcal{S}||\mathcal{A}|$ containing the square root inverse count of each state action transition. Considering the sampling error as shown in Eq. (14), we can derive the condition for $\hat{Q}^r$ converging to the lower-bound $Q^r$ at all $(s_t, a_t)$ through Eq. (12). The Proposition 3.6 provides the condition for $\hat{Q}^r$ to converge to the lower-bound of the $Q^r$. Proofs and discussions are in Appendix B.3.

**Proposition 3.6.** *For any $\pi_{\mathcal{M}}(a_t|s_t)$ with supp $\pi_{\mathcal{M}}(a_t|s_t) \subset \hat{\pi}_\beta$, $\forall s_t, a_t \in D$, the Q-value function $Q^r$ via iterating Eq. (12) satisfies:*

$$\hat{Q}^r(s_t, a_t) \leq Q^r(s_t, a_t) - \kappa \left[ (I - \gamma P^\pi)^{-1} \frac{\pi_{\mathcal{M}}}{\hat{\pi}_\beta} \right](s_t, a_t) + \left[ (I - \gamma P^\pi)^{-1} \frac{C_{r,p,\delta} R_{max}}{(1-\gamma)\sqrt{|\mathcal{D}|}} \right](s_t, a_t), \tag{15}$$

*Thus, if $\kappa \geq \frac{C_{r,p,\delta} R_{max}}{(1-\gamma)\sqrt{|\mathcal{D}(s_t,a_t)|}} \left[ \frac{\pi_{\mathcal{M}}(a_t|s_t)}{\hat{\pi}_\beta(a_t|s_t)} \right]^{-1}$, the iterative update Eq. (12) guarantees $\hat{Q}^r \leq Q^r$.*

Next, when $\pi_{\mathcal{M}}(a_t|s_t) = \pi_\theta(a_t|s_t)$, we obtain a not lower-bound for the Q-values estimates point-wise. based on Eq. (13). We abuse the notation $\frac{1}{\sqrt{|\mathcal{D}|}}$ to represent a vector of the inverse square root of state counts. Proofs and discussions are in Appendix B.4.

**Proposition 3.7.** *When $\pi_{\mathcal{M}}(a_t|s_t) = \pi_\theta(a_t|s_t)$, according to Eq. (13), we obtain a lower bound for the true value of $V^r = \mathbb{E}_{a_t \sim \pi_\theta(a_t|s_t)}[Q^r(s_t, a_t)]$ that satisfies the following inequality:*

$$\hat{V}^r(s_t) \leq V^r(s_t) - \kappa \left[ (I - \gamma P^\pi)^{-1} \mathbb{E}_{a_t \sim \pi_\theta} \left[ \frac{\pi_\theta}{\hat{\pi}_\beta} - 1 \right] \right](s_t) + \left[ (I - \gamma P^\pi)^{-1} \frac{C_{r,p,\delta} R_{max}}{(1-\gamma)\sqrt{|\mathcal{D}|}} \right](s_t), \tag{16}$$

*Thus, if $\kappa \geq \frac{C_{r,p,\delta} R_{max}}{(1-\gamma)\sqrt{|\mathcal{D}(s_t)|}} \left[ \mathbb{E}_{a_t \sim \pi_\theta} \left[ \frac{\pi_\theta(a_t|s_t)}{\pi_\beta(a_t|s_t)} - 1 \right] \right]^{-1}$, the Eq. (13) can guarantees $\hat{V}^r \leq V^r$.*

On the other hand, taking into account the need to satisfy the safety constraint $\mathbb{E}_{\rho(s_0)} \left[ \sum_a q(a_t|s_t) Q^{c_i}(s_t, a_t) \right] \leq \bar{c}_i$, the estimation of the cost Q-value must fulfill $\hat{Q}^{c_i} \geq Q^{c_i}$. Based on the above analysis we choose the penalty that maximizes the expected Q-value under special action-state transitions for unknown actions produced by $\pi_{\mathcal{R}}(a|s)$. Therefore, we define the Q-value of the iterative reward function as:

$$\hat{Q}^{c_i}_{k+1} \leftarrow \arg\min \frac{1}{2} \mathbb{E}_{s_t, a_t \sim D} \left[ \left[ (Q^{c_i}(s_t, a_t)) - \hat{\mathcal{B}}^\pi \hat{Q}^{c_i}_k(s_t, a_t) \right]^2 \right] - \chi \mathbb{E}_{\substack{s \sim D \\ a \sim \pi_{\mathcal{R}}(\cdot|s_t)}} \left[ Q^{c_i}(s_t, a_t) \right], \tag{17}$$

where $\chi$ is the trade-off factor. The $\pi_{\mathcal{R}}$ denotes the marginal distribution corresponding to the unknown action. The Proposition 3.8 provides an upper bound on the convergence of the fixed point of $\hat{Q}^{c_i}$, and clarifies the conditions for $\hat{Q}^{c_i}$ to converge to the upper bound of $Q^{c_i}$. Proofs and discussions are in Appendix B.5.

**Proposition 3.8.** *For any $\pi_{\mathcal{R}}(a_t|s_t)$ with supp $\pi_{\mathcal{R}}(a_t|s_t) \subset \hat{\pi}_\beta$, $\forall s_t, a_t \in D$, the Q-value function $Q^{c_i}$ via iterating Eq. (17) satisfies:*

$$\hat{Q}^{c_i}(s_t, a_t) \geq Q^{c_i}(s_t, a_t) + \chi \left[ (I - \gamma P^\pi)^{-1} \frac{\pi_{\mathcal{R}}}{\hat{\pi}_\beta} \right](s_t, a_t) - \left[ (I - \gamma P^\pi)^{-1} \frac{C_{r,p,\delta} C_{max}}{(1-\gamma)\sqrt{|\mathcal{D}|}} \right](s_t, a_t). \tag{18}$$

*Thus, if $\chi \geq \frac{C_{r,p,\delta} C_{max}}{(1-\gamma)\sqrt{|\mathcal{D}(s_t,a_t)|}} \left[ \frac{\pi_{\mathcal{R}}(a_t|s_t)}{\hat{\pi}_\beta(a_t|s_t)} \right]^{-1}$, the iterative update Eq. (17) can guarantee $\hat{Q}^{c_i} \geq Q^{c_i}$.*

### 3.2.2 Parametered Policy Update

After solving the optimal variational distribution of each state via Eq. (10), we need to obtain the policy parameters $\theta$. According to the solution target of Eq. (6), the optimization target can be obtained by eliminating the quantities irrelevant to $\theta$.

$$\mathcal{L}(\theta) = \max \mathbb{E}_{\tau \sim q} \left[ -\alpha D_{KL}(q(\cdot|s_t)||\pi_\theta(\cdot|s_t)) \right] = \max \alpha \mathbb{E}_{\rho(s_0)} \mathbb{E}_{q(a_t|s_t)} \left[ \log \pi_\theta(a_t|s_t) - \log q(a_t|s_t) \right], \tag{19}$$

where $\alpha \geq 0$ is the temperature parameters. In addition, the $q(a_t|s_t)$ is independent of $\pi_\theta(a_t|s_t)$. Therefore, the optimization objective of the above Eq. (19) can be rewriten as:

$$\mathcal{L}(\theta) = \max \mathbb{E}_{\rho(s_0)} \mathbb{E}_{q(a_t|s_t)} \big[ \log \pi_\theta(a_t|s_t) \big]. \tag{20}$$

## 4 Experimental Evaluation

In this section, we compare VOCE to previous offline safe RL methods in a range of domains and dataset compositions, including different action spaces and observation dimensions. To the best of my knowledge, there is currently no standardized dataset available in the field of offline safe RL. To facilitate further research and reproducibility of this work, we have collected a series of datasets using a trained online policy. The parameter settings of the dataset are in Appendix C.1.

### 4.1 Task and Baseline

**Task.** We choose *Point-Goal*, *Car-Goal*, *Point-Button* and *Car-Button* four tasks widely adopted in the field of safe RL [26, 29, 32–34], as the experimental tasks for this work. A detailed description of each task is in Appendix C.2.

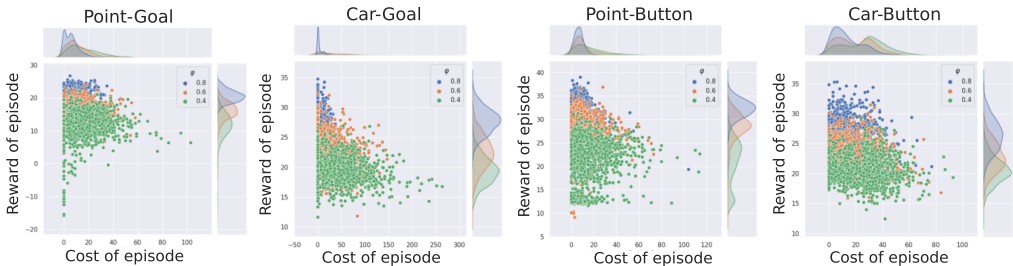

Figure 2: Distribution of rewards and costs for samples collected by different behavioral policies.

**Baselines.** BCQ-Lag is an offline safe RL algorithm that combines the Lagrange multiplier method with the BCQ [24] algorithm and employs adaptive penalty coefficients to implement offline constraints task. The C-CRR is an offline safe RL algorithm improved by the CRR [35] algorithm. It implements cost constraints by introducing cost evaluation Q-value function and Lagrange multipliers. Coptidice [10] is a policy optimization method based on the optimal stationary distribution space. The aforementioned three approaches are currently state-of-the-art algorithms in offline safe RL.

### 4.2 Performance Comparison Experiment

To evaluate the performance of VOCE on diverse task and behavior samples, we collected sample data from three distinct behaviors within four different tasks. We introduce parameter $\varphi$ to represent the proportion of trajectories in the sample data that satisfy the constraints. Then we employ $\varphi$ to characterize samples to different behaviors. Fig. 2 illustrates the marginal distribution of rewards and costs for sample trajectories in the dataset at different values of $\varphi$. The results from Fig. 2 reveal that as the value $\varphi$ increases, the mean cost decreases, and the mean reward increases. Fig. 3 displays the reward and cost curves of VOCE and the state-of-the-art offline safe RL methods under different $\varphi$ values for four tasks. The Fig. 3 results demonstrate that the VOCE achieves higher rewards across all tasks compared to the other three methods, while also satisfying or approaching the safety constraints threshold. Especially in the *Goal* task, VOCE consistently meets the safety constraints across different $\varphi$ values, while achieving higher reward returns. In the *Button* task, when the parameter $\varphi$ is small, VOCE struggles to ensure safety; however, the cost curve of VOCE remains lower than the other three methods. Based on the aforementioned results analysis, it can be concluded that the VOCE exhibits competitive performance across various combinations of samples from multiple tasks, particularly excelling in terms of safety compared to the state-of-the-art algorithms currently available.

### 4.3 Ablation Experiment

**The parameter $\varphi$ of the dataset.** Fig. 4 displays the boxplots of rewards and costs for the VOCE under different parameter values of $\varphi$ in the *Point-Goal* and *Point-Button* tasks. The results from

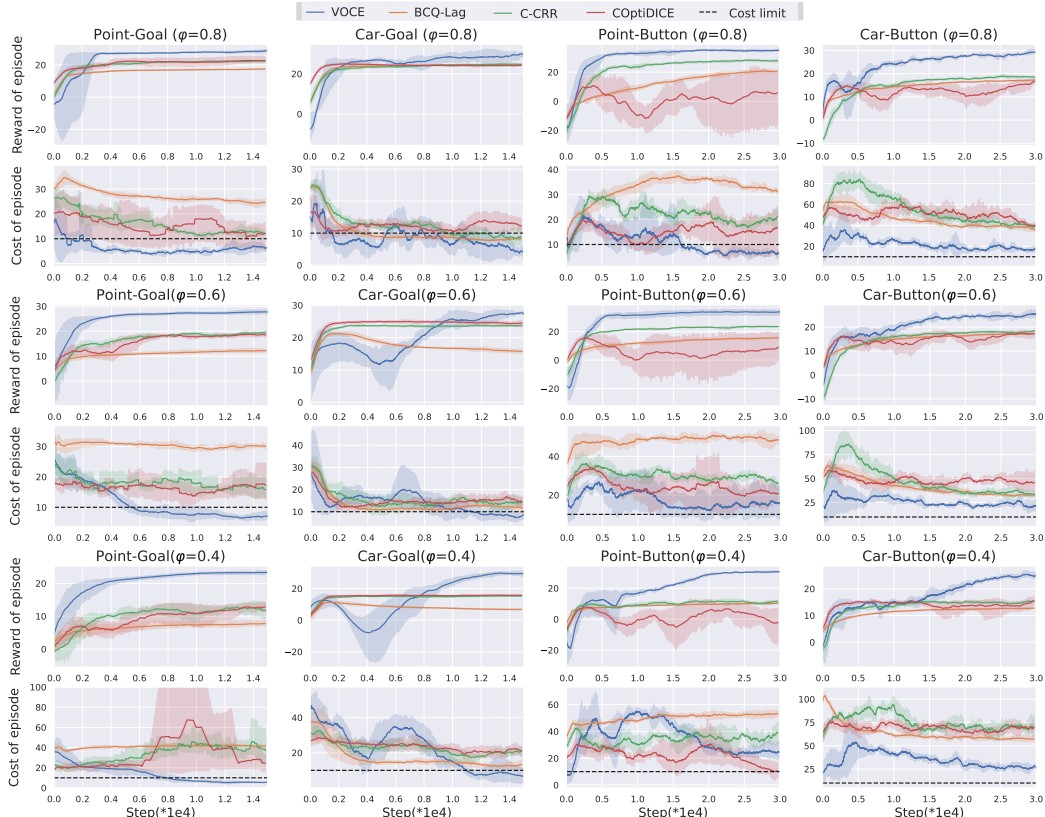

Figure 3: The reward and cost curves of VOCE and baseline algorithms with different sample data under 4 tasks. The curve is averaged over 3 random seeds, where the solid line is the mean and the shaded area is the standard deviation.

Fig. 4 reveals an intriguing phenomenon: VOCE does not achieve the highest reward and lowest cost at $\varphi$=1.0 but instead attains the highest reward within the range of 0.7 to 0.9 (The range will vary depending on changes in the sampling method or policy.). This indicates that appropriately increasing the number of constraint-satisfying trajectories in the dataset benefits VOCE in improving its rewards and reducing costs. However, excessive augmentation of constraint-satisfying trajectories in the dataset may lead to a decrease in rewards and even an increase in costs.

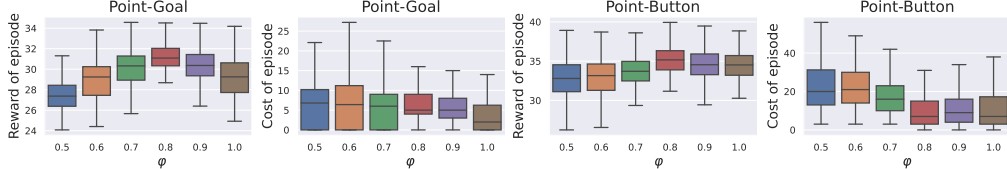

Figure 4: Ablation experiments with the same sampling policies and different $\varphi$ coefficients.

**Conservative estimation of rewards and costs.** To assess the impact of conservative estimation on VOCE, we conducted ablation experiments on the conservative estimation of both reward and cost Q-values. Fig. 5 illustrates the rewards and costs of VOCE, VOCE-Qr, and VOCE-Qc across four tasks. VOCE-Qr stands for the VOCE method with the conservative estimation of Q-values removed for rewards. VOCE-Qc represents the VOCE method with the conservative estimation of Q-values removed for costs. The results presented in Fig. 5 demonstrate that the rewards of VOCE-Qr are notably lower than those of VOCE. This indicates that employing a lower-bound conservative estimation of reward Q-values helps eliminate extrapolation errors caused by OOD actions, thereby significantly improving the reward of VOCE. Furthermore, the results from Fig. 5 reveal that the

rewards of VOCE-Qc are comparable to or even surpass those of VOCE. However, in some tasks, the costs of VOCE-Qc exceed the cost constraints. This suggests that utilizing an upper-bound conservative estimation of cost Q-values helps reduce the costs of VOCE, thereby enhancing the safety of VOCE.

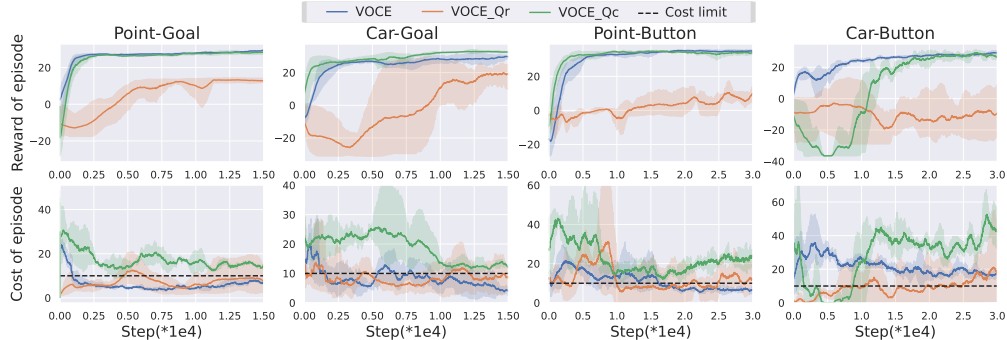

Figure 5: Ablation study for conservative Q-value estimation of reward and cost, and constrained regularize of the policy update. The curve is averaged over 3 random seeds, where the solid line is the mean and the shaded area is the standard deviation.

## 5   Related Work

In this section, we elaborate on the work pertaining to offline safe RL and scrutinize three aspects of safe RL, offline RL, and offline safe RL.

**Safe RL.** Currently, Safe RL typically addresses optimization objectives with constraints using the primal-dual framework. The Lagrange version of PPO and TRPO [36, 37] widely used as the underlying baselines combines the constraint information with the original objective function via Lagrange multipliers. CPO [38] is the first general-purpose policy search algorithm for safe RL that guarantees that the near constraints are satisfied at each iteration. However, the second-order method [38, 39] requires second-order information and thus brings a higher computational burden. To address the approximation errors associated with Taylor approximations and inverting a high-dimensional Fisher information matrix, first-order methods such as CUP [40] and APPO [41] achieve better performance and implementation.

**Offline RL.** Offline RL, also known as batch RL, considers the problem of updating a policy from an offline dataset without interacting with the environment. There are mainly two approaches of policy constraints and Q-value regularization to solve the problem of OOD actions in offline RL [42–44]. The earliest proposed Batch-Constrained deep Q-learning (BCQ) algorithm [24] is a typical policy-constrained offline RL algorithm, which employs CVAE to learn behavioral policy generation models, and finds the optimal policy by maximizing the Q-value. On this basis, a series of policy-constrained algorithms [45–48] are proposed, which mainly constrain the actions explored through the behavioral policy model. In addition, due to the obvious stability of the Q-value regularization approach, it has been extensively studied. A conservative Q-learning algorithm [30] proposed by Kumar et al., learns a conservative Q-value by enhancing the standard Bellman error objective with a simple Q-value regularizer. Subsequently, a series of Q-value regularization methods [49, 6, 50] were proposed which generally learn a conservative Q-value through regularization items or clipping Q-learning.

**Offline safe RL.** Offline safe RL is to solve the optimization problem of safety policy under the offline dataset. An offline safe RL algorithm with constraint penalty is proposed [16], which improves the objective function of the cost Q-value via using the constraint penalty item, and ensures the safety of the policy via terminating unsafe trajectory in advance. On the other hand, a constrained offline optimization algorithm (COPO) [11] defines the RL problem based on linear programming and constrains the distance between the final policy and offline sample behavior through regulation terms to solve the problem of offline safe RL algorithms. Additionally, this algorithm sets the discount factor $\gamma = 1$, and changes the maximizing discount reward objective to the maximizing mean reward objective. Similar to the COPO algorithm, an offline policy optimization via stationary distribution correction estimation (CoptiDICE) [18] also utilizes the linear programming method of RL to solve

the optimal stationary distribution instead of the gradient strategy, which expands the discount factor to $\gamma \in (0, 1]$.

## 6 Conclusion

In this work, we propose a variational optimization with conservative estimation for offline safe RL algorithm to address the problem of optimizing safe policies using offline datasets. We introduce probabilistic inference to reframe the problem of offline safe RL, and we mitigate estimation errors caused by parameter approximation through employing nonparametric variational distributions instead of parameterized policies. Furthermore, we utilize the upper and lower bounds of Q-values to estimate the Q-values of cost and reward, thereby mitigating extrapolation errors caused by OOD actions. Finally, extensive comparisons and ablation experiments demonstrate that the VOCE algorithm outperforms state-of-the-art algorithms in terms of safety.

## Acknowledgments and Disclosure of Funding

This work is supported by the National Natural Science Foundation of China (No. 62372329), in part by the National Key Research and Development Program of China (No. 2021YFB2501104), in part by Shanghai Rising Star Program (No.21QC1400900), in part by Tongji-Qomolo Autonomous Driving Commercial Vehicle Joint Lab Project, and in part by Xiaomi Young Talents Program. We thank Li Shen and Yiqin Yang for the insightful discussion.

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

# A Implementation Details

## A.1 Algorithm

The VOCE algorithm is shown in Algorithm 1.

---

**Algorithm 1:** VOCE Algorithm

---

**Input:** Dataset $\mathcal{D} = \{(s_t, a_t, r_t, s_{t+1})_{i=0}^n\}$
**Output:** Policy network $\pi_\theta$;
Q-value network of reward $Q^r, Q_t^r,$ ;
Q-value network of cost $Q^{c_i}, Q_t^{c_i}$;

1 **for** *each batch* **do**
2     Sample a batch of transitions $(s_t, a_t, r_t, s_{t+1})$ from the buffer $\mathcal{D}$;
3     Compute the Q-value $\hat{Q}^r(s_t, a_t)$ and $\hat{Q}^{c_i}(s_t, a_t)$ via Bellman backup;
4     $\hat{\mathcal{B}}^\pi \hat{Q}^r(s_t, a_t) = r(s_t, a_t) + \gamma \hat{Q}^r(s_{t+1}, a_{t+1})$;
5     $\hat{\mathcal{B}}^\pi \hat{Q}^{c_i}(s_t, a_t) = c_i(s_t, a_t) + \gamma \hat{Q}^{c_i}(s_{t+1}, a_{t+1})$;
6     Update the $Q^r$ via the Eq. (13);
7     Update the $Q^{c_i}$ via the Eq. (17);
8     Update the optimal dual variables $\eta$ and $\lambda$ via solving the convex optimization issue in the Eq. (11);
9     Compute the optimal variational distribution $q^*(a_t|s_t)$ by the Eq. (10) via $Q^r$, $Q^{c_i}$ and Lagrangian multipliers $\eta$ and $\lambda$ ;
10     Update the Policy Parameter $\pi_\theta$ via the Eq. (20);
11     $Q_t^r \leftarrow \tau Q^r + (1-\tau)Q_t^r$;
12     $Q_{t,}^{c_i} \leftarrow \tau Q^{c_i} + (1-\tau)Q_t^{c_i}$;

---

# B Proofs and Discussions

## B.1 Proof and Discussion the Proposition 3.4

**Proposition B.1.** *The closed-form solution of the optimal variational distribution that satisfies the safety constraint in Eq. (21) is given as:*

$$q^*(a_t|s_t) = \pi_\theta(a_t|s_t) \exp\left[\frac{Q^{rc}(s_t, a_t)}{\eta}\right] \exp\left(-\frac{\eta + \delta}{\eta}\right), \tag{21}$$

*where* $Q^{rc}(s_t, a_t) \triangleq Q^r(s_t, a_t) - \lambda^T Q^{c_i}(s_t, a_t)$.

***Proof.*** The unconstrained problem derived through the Lagrange multiplier method is expressed as follows:

$$\mathcal{L}(q, \lambda, \eta, \delta) = \max_{q \geq 0} \min_{\lambda \geq 0, \eta \geq 0, \delta} \mathbb{E}_{\rho(s_0)}\left[\sum_{a_t} q(a_t|s_t)Q^r(s_t, a_t) - \lambda^T\left[\sum_{a_t} Q^{c_i}(s_t, a_t) - \bar{c}_i\right]\right.$$
$$\left. - \eta\left[D_{KL}\left(q(a_t|s_t)||\pi_\theta(a_t|s_t)\right) - \epsilon\right] - \delta\left[\sum_{a_t} q(a_t|s_t) - 1\right]\right]. \tag{22}$$

To simplify Eq. (22), we introduce a composite Q-value and define it as $Q^{rc}(s_t, a_t) \triangleq Q^r(s_t, a_t) - \lambda^T Q^{c_i}(s_t, a_t)$. Then we rewrite the Eq. (22) as:

$$\mathcal{L}(q, \lambda, \eta, \delta) = \max_{q \geq 0} \min_{\lambda \geq 0, \eta \geq 0, \delta} \mathbb{E}_{\rho(s_0)}\left[\sum_{a_t} q(a_t|s_t)Q^{rc}(s_t, a_t) - \eta \sum_{a_t} q(a_t|s_t) \log \frac{q(a_t|s_t)}{\pi_\theta(a_t|s_t)}\right.$$
$$\left. - \delta \sum_{a_t} q(a_t|s_t) + \lambda^T \bar{c}_i + \eta\epsilon + \delta\right]. \tag{23}$$

For a fixed $\lambda$ and $\eta$, we write the dual problem [17, 18] of the above optimization problem of maximization $\max_{q \geq 0} \mathcal{L}(q, \lambda, \eta)$:

$$\max_{q \geq 0} \min_{u \geq 0} \mathcal{L}(q, \lambda, \eta) + \sum_{s_t, a_t} \mu(s_t, a_t)q(a_t|s_t). \tag{24}$$

Since the strong duality holds, satisfying the KKT condition is a necessary and sufficient condition for the solution of the primal and dual problems.

***Condition 1 (Primal feasibility).*** $q^*(a_t|s_t) \geq 0, \forall s_t, a_t$

***Condition 2 (Dual feasibility).*** $\mu^*(s_t, a_t) \geq 0, \forall s_t, a_t$

***Condition 3 (Stationarity).*** $Q^r(s_t, a_t) - \lambda^T Q^{c_i}(s_t, a_t) - \eta - \eta \log q^*(a_t|s_t) - \eta \log \pi_\theta(a_t|s_t) + \mu^*(s_t, a_t) = 0, \forall s_t, a_t$

***Condition 4 (Complementary).*** $q^*(a_t|s_t)\mu^*(s_t, a_t) = 0, \forall s_t, a_t$

Form **Stationarity** we have:

$$q^*(a_t|s_t) = \pi_\theta(a_t|s_t) \exp\left[\frac{Q^{rc}(s_t, a_t)}{\eta}\right] \exp\left(-\frac{\eta + \delta - u^*(s_t, a_t)}{\eta}\right). \tag{25}$$

Since the $\pi_\theta(a_t|s_t) > 0$, thus the $q^*(a_t|s_t) > 0$. Then the **Primal feasibility** is always satisfied. Furthermore, we only need to consider the cases where $q^*(a_t|s_t) > 0$.

(**Case 1** $q^*(a_t|s_t) > 0$ ) In this case, $\mu^*(s_t, a_t) = 0$ where Dual feasibility and Complementary slackness are satisfied. In addition, Stationarity also applies to :

$$q^*(a_t|s_t) = \pi_\theta(a_t|s_t) \exp\left[\frac{Q^{rc}(s_t, a_t)}{\eta}\right] \exp\left(-\frac{\eta + \delta}{\eta}\right). \tag{26}$$

Therefore, KKT conditions (conditions 1-4 ) are satisfied.

## B.2 Proof and Discussion the Proposition 3.5

**Proposition B.2.** *The $\lambda, \eta$ are dual variables and are solved via the convex optimization problem of the following formula.*

$$\mathcal{L}(\lambda, \eta) = \min_{\lambda \geq 0, \eta \geq 0} \mathbb{E}_{\rho(s_0)}\left[\eta \log \sum_{a_t} \pi_\theta(a_t|s_t) \exp\left(\frac{Q^{rc}(s_t, a_t)}{\eta}\right) + \lambda^T \bar{c}_i + \eta\epsilon\right]. \tag{27}$$

*Proof.* By rearranging terms and taking the expectation over $a_t$, Eq. (21) can be rewritten as:

$$\exp\left(\frac{\eta + \delta}{\eta}\right) \sum_{a_t} q^*(a_t|s_t) = \sum_{a_t} \pi_\theta(a_t|s_t) \exp\left[\frac{Q^{rc}(s_t, a_t)}{\eta}\right], \tag{28}$$

where the $\sum_{a_t} q^*(a_t|s_t) = 1$, thus the Eq. (28) can be rewritten as:

$$\frac{\eta + \delta}{\eta} = \log \sum_{a_t} \pi_\theta(a_t|s_t) \exp\left[\frac{Q^{rc}(s_t, a_t)}{\eta}\right]. \tag{29}$$

By substituting Eq. (29) and (26) into Eq. (22) to eliminate $q(a_t|s_t)$, we can rewrite Eq. (22) as:

$$
\begin{aligned}
\mathcal{L}(\lambda, \eta, \delta) &= \min_{\lambda \geq 0, \eta \geq 0, \delta} \mathbb{E}_{\rho(s_0)}\left[\sum_{a_t} q^*(a_t|s_t)Q^{rc}(s_t, a_t) - \eta \sum_{a_t} q^*(a_t|s_t) \log \frac{q^*(a|s)}{\pi_\theta(a|s)}\right.\\
&\qquad\qquad\qquad\left. - \delta \sum_{a_t} q^*(a_t|s_t) + \lambda^T \bar{c}_i + \eta\epsilon + \delta\right]\\
&= \min_{\lambda \geq 0, \eta \geq 0, \delta} \mathbb{E}_{\rho(s_0)}\left[\sum_{a_t} q^*(a_t|s_t)Q^{rc}(s_t, a_t) - \eta \sum_{a_t} q^*(a_t|s_t)\left[\frac{Q^{rc}(s_t, a_t) - \eta - \delta}{\eta}\right]\right.\\
&\qquad\qquad\qquad\left. - \delta \sum_{a_t} q^*(a_t|s_t) + \lambda^T \bar{c}_i + \eta\epsilon + \delta\right]\\
&= \min_{\lambda \geq 0, \eta \geq 0, \delta} \mathbb{E}_{\rho(s_0)}\left[\sum_{a_t} q^*(a_t|s_t)\left[\eta + \delta\right] - \delta \sum_{a_t} q^*(a_t|s_t) + \lambda^T \bar{c}_i + \eta\epsilon + \delta\right]\\
&= \min_{\lambda \geq 0, \eta \geq 0, \delta} \mathbb{E}_{\rho(s_0)}\left[\eta \log \sum_{a_t} \pi_\theta(a_t|s_t) \exp\left[\frac{Q^{rc}(s_t, a_t)}{\eta}\right] + \lambda^T \bar{c}_i + \eta\epsilon\right].
\end{aligned}
\tag{30}
$$

Since Eq. (30) eliminates $\delta$, the dual optimization problem of $\lambda$ and $\eta$ can be written as:

$$\mathcal{L}(\lambda, \eta) = \min_{\lambda \geq 0, \eta \geq 0} \mathbb{E}_{\rho(s_0)}\left[\eta \log \sum_{a_t} \pi_\theta(a_t|s_t) \exp\left[\frac{Q^{rc}(s_t, a_t)}{\eta}\right] + \lambda^T \bar{c}_i + \eta\epsilon\right]. \tag{31}$$

## B.3  Proof and Discussion the Proposition 3.6

**Proposition B.3.** *For any $\pi_{\mathcal{M}}(a_t|s_t)$ with supp $\pi_{\mathcal{M}}(a_t|s_t) \subset \hat{\pi}_{\beta}$, $\forall s_t, a_t \in D$, the Q-value function $Q^r$ via iterating Eq. (12) satisfies:*

$$\hat{Q}^r(s_t, a_t) \leq Q^r(s_t, a_t) - \kappa \left[ (I - \gamma P^{\pi})^{-1} \frac{\pi_{\mathcal{M}}}{\hat{\pi}_{\beta}} \right](s_t, a_t) + \left[ (I - \gamma P^{\pi})^{-1} \frac{C_{r,p,\delta} R_{max}}{(1-\gamma)\sqrt{|\mathcal{D}|}} \right](s_t, a_t). \quad (32)$$

*Thus, if $\kappa \geq \frac{C_{r,p,\delta} R_{max}}{(1-\gamma)\sqrt{|\mathcal{D}(s_t,a_t)|}} \left[ \frac{\pi_{\mathcal{M}}(a_t|s_t)}{\pi_{\beta}(a_t|s_t)} \right]^{-1}$, the iterative update Eq. (12) can guarantees $\hat{Q}^r \leq Q^r$.*

***Proof.*** The iterative equation for updating the reward Q-value, which involves an additional minimization of the Q-value and the standard Bellman error objective, is expressed as follows:

$$\hat{Q}^r_{k+1} \leftarrow \arg\min \frac{1}{2} \mathbb{E}_{s_t,a_t \sim D} \left[ \left[ (Q^r(s_t, a_t)) - \hat{\mathcal{B}}^{\pi} \hat{Q}^r_k(s_t, a_t) \right]^2 \right] +$$
$$\kappa \mathbb{E}_{s_t \sim D, a_t \sim \pi_{\mathcal{M}}(\cdot|s_t)} \left[ Q^r(s_t, a_t) \right], \quad (33)$$

where the $\hat{\mathcal{B}}^{\pi} \hat{Q}^r(s_t, a_t) = r(s_t, a_t) + \gamma \hat{Q}^r(s_{t+1}, a_{t+1})$. With importance sampling, the Eq. (33) can be written as:

$$\hat{Q}^r_{k+1} \leftarrow \arg\min_{Q^r} \frac{1}{2} \mathbb{E}_{s_t,a_t \sim D} \left[ \left[ (Q^r(s_t, a_t)) - \hat{\mathcal{B}}^{\pi} \hat{Q}^r_k(s, a) \right]^2 \right] +$$
$$\kappa \mathbb{E}_{s_t \sim D, a_t \sim \hat{\pi}_{\beta}(\cdot|s_t)} \left[ \frac{\pi_{\mathcal{M}}(a_t|s_t)}{\hat{\pi}_{\beta}(a_t|s_t)} Q^r(s_t, a_t) \right]. \quad (34)$$

Setting the derivative of Eq. (34) to 0, we obtain the following expression:

$$\hat{Q}^r_{k+1}(s_t, a_t) = \hat{\mathcal{B}}^{\pi} \hat{Q}_k(s_t, a_t) - \kappa \frac{\pi_{\mathcal{M}}(a_t|s_t)}{\hat{\pi}_{\beta}(a_t|s_t)}. \quad (35)$$

Note that the Eq. (34) utilizes the empirical Bellman operator $\hat{\mathcal{B}}^{\pi}$ instead of the actual Bellman Operator $\mathcal{B}^{\pi}$. Following the related work [23, 31], we employ the concentration properties to determine the error. For any $\forall s_t, a_t \in D$, with the probability $\leq 1 - \delta$, the error can be written as:

$$|\mathcal{B}^{\pi} \hat{Q}(s_t, a_t) - \hat{\mathcal{B}}^{\pi} \hat{Q}(s_t, a_t)| \leq \frac{C_{r,p,\delta} R_{max}}{(1-\gamma)\sqrt{|\mathcal{D}(s_t, a_t)|}}, \quad (36)$$

where $C_{r,p,\delta}$ is a constant depending on the concentration properties of $r(s_t, a_t)$, $p(s_{t+1}|s_t, a_t)$ and the $\delta, \delta \in (0, 1)$. The $(\mathcal{D}(s_t, a_t))^{-\frac{1}{2}}$ represents a vector of size $|\mathcal{S}||\mathcal{A}|$ containing the square root inverse count of each state action transition. $R_{max} = \max_{s_t, a_t \sim D} \{r(s_t, a_t)\}$ represents the maximum reward obtained in a single step. Based on the inequality of Eq. (36) and Eq. (35) we obtain:

$$\hat{Q}^r_{k+1}(s_t, a_t) = \hat{\mathcal{B}}^{\pi} \hat{Q}^r_k(s_t, a_t) - \kappa \frac{\pi_{\mathcal{M}}(a_t|s_t)}{\hat{\pi}_{\beta}(a_t|s_t)}$$
$$\leq \mathcal{B}^{\pi} \hat{Q}^r_k(s_t, a_t) + \frac{C_{r,p,\delta} R_{max}}{(1-\gamma)\sqrt{|\mathcal{D}(s_t, a_t)|}} - \kappa \frac{\pi_{\mathcal{M}}(a_t|s_t)}{\hat{\pi}_{\beta}(a_t|s_t)}. \quad (37)$$

The fixed point in the updating process of Eq. (37) can be expressed as:

$$\hat{Q}^r(s_t, a_t) \leq \mathcal{B}^{\pi} \hat{Q}^r(s_t, a_t) + \frac{C_{r,p,\delta} R_{max}}{(1-\gamma)\sqrt{|\mathcal{D}(s_t, a_t)|}} - \kappa \frac{\pi_{\mathcal{M}}(a_t|s_t)}{\hat{\pi}_{\beta}(a_t|s_t)}$$
$$\leq r + \gamma P^{\pi} \hat{Q}^r(s_t, a_t) + \frac{C_{r,p,\delta} R_{max}}{(1-\gamma)\sqrt{|\mathcal{D}(s_t, a_t)|}} - \kappa \frac{\pi_{\mathcal{M}}(a_t|s_t)}{\hat{\pi}_{\beta}(a_t|s_t)}$$
$$\leq Q^r(s_t, a_t) - \gamma P^{\pi} Q^r(s_t, a_t) + \gamma P^{\pi} \hat{Q}^r(s_t, a_t) + \frac{C_{r,p,\delta} R_{max}}{(1-\gamma)\sqrt{|\mathcal{D}(s_t, a_t)|}} - \kappa \frac{\pi_{\mathcal{M}}(a_t|s_t)}{\hat{\pi}_{\beta}(a_t|s_t)}, \quad (38)$$

$$(I - \gamma P^{\pi}) \hat{Q}^r(s_t, a_t) \leq (I - \gamma P^{\pi}) Q^r(s_t, a_t) + \frac{C_{r,p,\delta} R_{max}}{(1-\gamma)\sqrt{|\mathcal{D}(s_t, a_t)|}} - \kappa \frac{\pi_{\mathcal{M}}(a_t|s_t)}{\hat{\pi}_{\beta}(a_t|s_t)}, \quad (39)$$

$$\hat{Q}^r(s_t, a_t) \le Q^r(s_t, a_t) + (I - \gamma P^\pi)^{-1} \left[ \frac{C_{r,p,\delta} R_{max}}{(1-\gamma)\sqrt{|\mathcal{D}(s_t, a_t)|}} - \kappa \frac{\pi_\mathcal{M}(a_t|s_t)}{\hat{\pi}_\beta(a_t|s_t)} \right]$$

$$\le Q^r(s_t, a_t) - \kappa(I - \gamma P^\pi)^{-1} \frac{\pi_\mathcal{M}(a_t|s_t)}{\hat{\pi}_\beta(a_t|s_t)} + (I - \gamma P^\pi)^{-1} \frac{C_{r,p,\delta} R_{max}}{(1-\gamma)\sqrt{|\mathcal{D}(s_t, a_t)|}}. \tag{40}$$

To guarantee the lower bound of $\hat{Q}^r$, we eliminate the potential overestimation caused by $\frac{C_{r,p,\delta} R_{max}}{(1-\gamma)\sqrt{|\mathcal{D}(s_t,a_t)|}}$ via choosing an appropriate $\kappa$. Since $(I - \gamma P^\pi)^{-1}$ is a matrix of non-negative entries, the $\kappa$ that guarantees the lower bound of $Q^r$ needs to satisfy:

$$\kappa \ge \frac{C_{r,p,\delta} R_{max}}{(1-\gamma)\sqrt{|\mathcal{D}(s_t, a_t)|}} \left[ \frac{\pi_\mathcal{M}(a_t|s_t)}{\hat{\pi}_\beta(a_t|s_t)} \right]^{-1}. \tag{41}$$

## B.4    Proof and Discussion the Proposition 3.7

**Proposition B.4.** *When $\pi_\mathcal{M}(a_t|s_t) = \pi_\theta(a_t|s_t)$, according to Eq. (13), we obtain a lower bound for the true value of $V^r = \mathbb{E}_{a_t \sim \pi_\theta(a_t|s_t)}[Q^r(s_t, a_t)]$ that satisfies the following inequality:*

$$\hat{V}^r(s_t) \le V^r(s_t) - \kappa \left[ (I - \gamma P^\pi)^{-1} \mathbb{E}_{a_t \sim \pi_\theta} \left[ \frac{\pi_\theta(a_t|s_t)}{\hat{\pi}_\beta(a_t|s_t)} - 1 \right] \right] (s_t) + \left[ (I - \gamma P^\pi)^{-1} \frac{C_{r,p,\delta} R_{max}}{(1-\gamma)\sqrt{|\mathcal{D}|}} \right] (s_t). \tag{42}$$

*Thus, if $\kappa \ge \frac{C_{r,p,\delta} R_{max}}{(1-\gamma)\sqrt{|\mathcal{D}(s_t)|}} \left[ \mathbb{E}_{a_t \sim \hat{\pi}_\beta} \left[ \frac{\pi_\theta(a_t|s_t)}{\pi_\theta(a_t|s_t)} - 1 \right] \right]^{-1}$, the Eq. (13) can guarantee $\hat{V}^r \le V^r$.*

***Proof.*** The following Eq. (43) represents the iterative formula for the reward Q-value with an additional maximization term introduced under the sample distribution $\hat{\pi}_\beta$.

$$\hat{Q}^r_{k+1} \leftarrow \arg\min \frac{1}{2} \mathbb{E}_{s_t, a_t \sim D} \left[ \left[ (Q^r(s_t, a_t)) - \hat{\mathcal{B}}^\pi \hat{Q}^r_k(s_t, a_t) \right]^2 \right] +$$

$$\kappa \left[ \mathbb{E}_{\substack{s_t \sim D \\ a_t \sim \pi_\mathcal{M}(\cdot|s_t)}} Q^r(s_t, a_t) - \mathbb{E}_{\substack{s_t \sim D \\ a_t \sim \hat{\pi}_\beta(\cdot|s_t)}} Q^r(s_t, a_t) \right], \tag{43}$$

where $\kappa$ is a tradeoff factor. With importance sampling, the Eq. (43) can be rewritten as:

$$\hat{Q}^r_{k+1} \leftarrow \arg\min \frac{1}{2} \mathbb{E}_{s_t, a_t \sim D} \left[ \left[ (Q^r(s_t, a_t)) - \hat{\mathcal{B}}^\pi \hat{Q}^r_k(s_t, a_t) \right]^2 \right] +$$

$$\kappa \left[ \mathbb{E}_{\substack{s_t \sim D \\ a_t \sim \hat{\pi}_\beta(\cdot|s_t)}} \frac{\pi_\mathcal{M}(a_t|s_t)}{\hat{\pi}_\beta(a_t|s_t)} Q^r(s_t, a_t) - \mathbb{E}_{\substack{s_t \sim D \\ a_t \sim \hat{\pi}_\beta(\cdot|s_t)}} Q^r(s_t, a_t) \right]. \tag{44}$$

Setting the derivative of Eq. (44) to 0, we obtain the following expression:

$$\hat{Q}^r_{k+1}(s_t, a_t) = \hat{\mathcal{B}}^\pi \hat{Q}_k(s_t, a_t) - \kappa \left[ \frac{\pi_\mathcal{M}(a_t|s_t)}{\hat{\pi}_\beta(a_t|s_t)} - 1 \right] \tag{45}$$

By taking the expectation of both side of Eq. (45) with respect of the action, we obtain:

$$\hat{V}^r_{k+1}(s_t) = \hat{\mathcal{B}}^\pi \hat{V}_k(s_t) - \kappa \sum_{a_t} \pi_\theta(a_t|s_t) \left[ \frac{\pi_\mathcal{M}(a_t|s_t)}{\hat{\pi}_\beta(a_t|s_t)} - 1 \right]$$

$$= \hat{\mathcal{B}}^\pi \hat{V}_k(s_t) - \kappa \sum_{a_t} \left[ \pi_\theta(a_t|s_t) - \hat{\pi}_\beta(a_t|s_t) + \hat{\pi}_\beta(a_t|s_t) \right] \left[ \frac{\pi_\mathcal{M}(a_t|s_t)}{\hat{\pi}_\beta(a_t|s_t)} - 1 \right]. \tag{46}$$

When $\pi_\mathcal{M}(a_t|s_t) = \pi_\theta(a_t|s_t)$, we obtain from Eq. (46):

$$\hat{V}^r_{k+1}(s_t) = \hat{\mathcal{B}}^\pi \hat{V}_k(s_t) - \kappa \underbrace{\sum_{a_t} \left[ \frac{(\pi_\theta(a_t|s_t) - \hat{\pi}_\beta(a_t|s_t))^2}{\hat{\pi}_\beta(a_t|s_t)} \right]}_{\ge 0} - \kappa \left( \underbrace{\sum_{a_t} \pi_\theta(a_t|s_t)}_{=1} - \underbrace{\sum_{a_t} \hat{\pi}_\beta(a_t|s_t)}_{=1} \right). \tag{47}$$

Since $\sum_{a_t} \pi_\theta(a_t|s_t) = \sum_{a_t} \hat{\pi}_\beta(a_t|s_t) = 1$, we can conclude that the value estimate $\hat{V}_k(s_t)$ is underestimated as $\hat{V}_{k+1}^r(s_t) \leq \hat{\mathcal{B}}^\pi \hat{V}_k(s_t)$ based on Eq. (47). Next, considering the sampling error, we obtain from Eq. (45) and Eq. 36:

$$\hat{Q}_{k+1}^r(s_t, a_t) = \hat{\mathcal{B}}^\pi \hat{Q}_k^r(s_t, a_t) - \kappa \left[ \frac{\pi_\mathcal{M}(a_t|s_t)}{\hat{\pi}_\beta(a_t|s_t)} - 1 \right]$$

$$\leq \mathcal{B}^\pi \hat{Q}_k(s_t, a_t) - \kappa \left[ \frac{\pi_\mathcal{M}(a_t|s_t)}{\hat{\pi}_\beta(a_t|s_t)} - 1 \right] + \frac{C_{r,p,\delta} R_{max}}{(1-\gamma)\sqrt{|\mathcal{D}(s_t, a_t)|}}. \qquad (48)$$

When $\pi_\mathcal{M}(a_t|s_t) = \pi_\theta(a_t|s_t)$, the fixed point of the update process in Eq. 48 can be expressed as:

$$\hat{Q}^r(s_t, a_t) \leq \mathcal{B}^\pi \hat{Q}^r(s_t, a_t) - \kappa \left[ \frac{\pi_\theta(a_t|s_t)}{\hat{\pi}_\beta(a_t|s_t)} - 1 \right] + \frac{C_{r,p,\delta} R_{max}}{(1-\gamma)\sqrt{|\mathcal{D}(s_t, a_t)|}}$$

$$\leq r + \gamma P^\pi \hat{Q}^r(s_t, a_t) - \kappa \left[ \frac{\pi_\theta(a_t|s_t)}{\hat{\pi}_\beta(a_t|s_t)} - 1 \right] + \frac{C_{r,p,\delta} R_{max}}{(1-\gamma)\sqrt{|\mathcal{D}(s_t, a_t)|}}$$

$$\leq Q^r(s_t, a_t) - \gamma P^\pi Q^r(s_t, a_t) + \gamma P^\pi \hat{Q}^r(s_t, a_t) - \kappa \left[ \frac{\pi_\theta(a_t|s_t)}{\hat{\pi}_\beta(a_t|s_t)} - 1 \right] + \frac{C_{r,p,\delta} R_{max}}{(1-\gamma)\sqrt{|\mathcal{D}(s_t, a_t)|}}$$
$$(49)$$

$$(I - \gamma P^\pi)\hat{Q}^r(s_t, a_t) \leq (I - \gamma P^\pi)Q^r(s_t, a_t) - \kappa \left[ \frac{\pi_\theta(a_t|s_t)}{\hat{\pi}_\beta(a_t|s_t)} - 1 \right] + \frac{C_{r,p,\delta} R_{max}}{(1-\gamma)\sqrt{|\mathcal{D}(s_t, a_t)|}}, \qquad (50)$$

$$\hat{Q}^r(s_t, a_t) \leq Q^r(s_t, a_t) - \kappa(I - \gamma P^\pi)^{-1} \left[ \frac{\pi_\theta(a_t|s_t)}{\hat{\pi}_\beta(a_t|s_t)} - 1 \right] + (I - \gamma P^\pi)^{-1} \frac{C_{r,p,\delta} R_{max}}{(1-\gamma)\sqrt{|\mathcal{D}(s_t, a_t)|}}, \qquad (51)$$

$$\mathbb{E}_{a_t \sim \pi_\theta} \hat{Q}^r(s_t, a_t) \leq \mathbb{E}_{a_t \sim \pi_\theta} Q^r(s_t, a_t) - \kappa(I - \gamma P^\pi)^{-1} \mathbb{E}_{a_t \sim \pi_\theta} \left[ \frac{\pi_\theta(a_t|s_t)}{\hat{\pi}_\beta(a_t|s_t)} - 1 \right]$$

$$+ (I - \gamma P^\pi)^{-1} \mathbb{E}_{a_t \sim \pi_\theta} \frac{C_{r,p,\delta} R_{max}}{(1-\gamma)\sqrt{|\mathcal{D}(s_t, a_t)|}}, \qquad (52)$$

$$\hat{V}^r(s_t) \leq V^r(s_t) - \kappa(I - \gamma P^\pi)^{-1} \mathbb{E}_{a_t \sim \pi_\theta} \left[ \frac{\pi_\mathcal{M}(a_t|s_t)}{\pi_\beta(a_t|s_t)} - 1 \right] + (I - \gamma P^\pi)^{-1} \frac{C_{r,p,\delta} R_{max}}{(1-\gamma)\sqrt{|\mathcal{D}(s_t)|}}. \qquad (53)$$

To guarantee the lower bound of $\hat{V}^r$, we eliminate the potential overestimation caused by $\frac{C_{r,p,\delta} R_{max}}{(1-\gamma)\sqrt{|\mathcal{D}(s_t)|}}$ via choosing an appropriate $\kappa$. Since $(I - \gamma P^\pi)^{-1}$ is a matrix of non-negative entries, the $\kappa$ that guarantees the lower bound of $V^r$ needs to satisfy:

$$\kappa \geq \frac{C_{r,p,\delta} R_{max}}{(1-\gamma)\sqrt{|\mathcal{D}(s_t)|}} \left[ \mathbb{E}_{a_t \sim \pi_\theta} \left[ \frac{\pi_\theta(a_t|s_t)}{\hat{\pi}_\beta(a_t|s_t)} - 1 \right] \right]^{-1}. \qquad (54)$$

### B.5   Proof and Discussion the Proposition 3.8

**Proposition B.5.** *For any $\pi_\mathcal{R}(a_t|s_t)$ with supp $\pi_\mathcal{R}(a_t|s_t) \subset \hat{\pi}_\beta$, $\forall s_t, a_t \in D$, the Q-value function $Q^{c_i}$ via iterating Eq. (17) satisfies:*

$$\hat{Q}^{c_i}(s_t, a_t) \geq Q^{c_i}(s_t, a_t) + \chi \left[ (I - \gamma P^\pi)^{-1} \frac{\pi_\mathcal{R}}{\hat{\pi}_\beta} \right](s_t, a_t) - \left[ (I - \gamma P^\pi)^{-1} \frac{C_{r,p,\delta} C_{max}}{(1-\gamma)\sqrt{|\mathcal{D}|}} \right](s_t, a_t). \qquad (55)$$

*Thus, if $\chi \geq \frac{C_{r,p,\delta} C_{max}}{(1-\gamma)\sqrt{|\mathcal{D}(s_t, a_t)|}} \left[ \frac{\pi_\mathcal{R}(a_t|s_t)}{\hat{\pi}_\beta(a_t|s_t)} \right]^{-1}$, the iterative update Eq. (17) can guarantee $\hat{Q}^{c_i} \geq Q^{c_i}$.*

**Proof.** Using the importance sampling, Eq. (17) can be written as:

$$\hat{Q}_{k+1}^{c_i}(s_t, a_t) \leftarrow \arg\min_{Q^{c_i}} \frac{1}{2} \mathbb{E}_{s_t, a_t \sim D} \left[ \left[ (Q^{c_t}(s_t, a_t)) - \hat{\mathcal{B}}^\pi \hat{Q}_k^{c_i}(s_t, a_t) \right]^2 \right] -$$

$$\chi \mathbb{E}_{s_t \sim D, a_t \sim \hat{\pi}_\beta(\cdot|s_t)} \left[ \frac{\pi_\mathcal{R}(a|s)}{\hat{\pi}_\beta(a_t|s_t)} Q^{c_i}(s_t, a_t) \right]. \qquad (56)$$

Setting the derivative of Eq. (56) to 0, we obtain the following expression:

$$\hat{Q}_{k+1}^{c_i}(s_t, a_t) = \hat{\mathcal{B}}^\pi \hat{Q}_k^{c_i}(s, a) + \chi \frac{\pi_\mathcal{R}(a_t|s_t)}{\hat{\pi}_\beta(a_t|s_t)} \tag{57}$$

Note that the Eq. (57) utilizes the empirical Bellman operator $\hat{\mathcal{B}}^\pi$ instead of the actual Bellman Operator $\mathcal{B}^\pi$. Following the related work [23, 31], we employ the concentration properties to determine the error. For any $\forall s_t, a_t \in D$, with the probability $\leq 1 - \delta$, the error can be written as:

$$|\mathcal{B}^\pi \hat{Q}(s_t, a_t) - \hat{\mathcal{B}}^\pi \hat{Q}(s_t, a_t)| \leq \frac{C_{r,p,\delta} C_{max}}{(1-\gamma)\sqrt{|\mathcal{D}(s_t, a_t)|}}. \tag{58}$$

where $C_{r,p,\delta}$ is a constant depending on the concentration properties of $c(s_t, a_t)$, $p(s_{t+1}|s_t, a_t)$ and the $\delta, \delta \in (0,1)$. The $(\mathcal{D}(s_t, a_t))^{-\frac{1}{2}}$ represents a vector of size $|\mathcal{S}||\mathcal{A}|$ containing the square root inverse count of each state action transition. $C_{max} = \max_{s_t, a_t \sim D}\{c_i(s_t, a_t)\}$ represents the maximum cost obtained in a single step. Based on the inequality of Eq. (58) and Eq. (57) we can obtain:

$$\begin{aligned}
\hat{Q}_{k+1}^{c_i}(s_t, a_t) &= \hat{\mathcal{B}}^\pi \hat{Q}_k^{c_i}(s_t, a_t) + \chi \frac{\pi_\mathcal{R}(a_t|s_t)}{\hat{\pi}_\beta(a_t|s_t)} \\
&\geq \mathcal{B}^\pi \hat{Q}_k^{c_i}(s_t, a_t) - \frac{C_{r,p,\delta} C_{max}}{(1-\gamma)\sqrt{|\mathcal{D}(s_t, a_t)|}} + \kappa \frac{\pi_\mathcal{R}(a_t|s_t)}{\hat{\pi}_\beta(a_t|s_t)}.
\end{aligned} \tag{59}$$

The fixed point in the updating process of Eq. (59) can be expressed as:

$$\begin{aligned}
\hat{Q}^{c_i}(s_t, a_t) &\geq \mathcal{B}^\pi \hat{Q}^{c_i}(s_t, a_t) - \frac{C_{r,p,\delta} C_{max}}{(1-\gamma)\sqrt{|\mathcal{D}(s_t, a_t)|}} + \chi \frac{\pi_\mathcal{R}(a_t|s_t)}{\hat{\pi}_\beta(a_t|s_t)} \\
&\geq c + \gamma P^\pi \hat{Q}^{c_i}(s_t, a_t) - \frac{C_{r,p,\delta} C_{max}}{(1-\gamma)\sqrt{|\mathcal{D}(s_t, a_t)|}} + \chi \frac{\pi_\mathcal{R}(a_t|s_t)}{\hat{\pi}_\beta(a_t|s_t)} \\
&\geq Q^{c_i}(s_t, a_t) - \gamma P^\pi Q^{c_i}(s_t, a_t) + \gamma P^\pi \hat{Q}^{c_i}(s_t, a_t) - \frac{C_{r,p,\delta} C_{max}}{(1-\gamma)\sqrt{|\mathcal{D}(s_t, a_t)|}} + \chi \frac{\pi_\mathcal{R}(a_t|s_t)}{\hat{\pi}_\beta(a_t|s_t)},
\end{aligned} \tag{60}$$

$$(I - \gamma P^\pi)\hat{Q}^{c_i}(s_t, a_t) \geq (I - \gamma P^\pi)Q^{c_i}(s_t, a_t) - \frac{C_{r,p,\delta} C_{max}}{(1-\gamma)\sqrt{|\mathcal{D}(s_t, a_t)|}} + \chi \frac{\pi_\mathcal{R}(a_t|s_t)}{\hat{\pi}_\beta(a_t|s_t)}, \tag{61}$$

$$\begin{aligned}
\hat{Q}^{c_i}(s_t, a_t) &\geq Q^{c_i}(s_t, a_t) - (I - \gamma P^\pi)^{-1}\left[\frac{C_{r,p,\delta} C_{max}}{(1-\gamma)\sqrt{|\mathcal{D}(s_t, a_t)|}} + \chi \frac{\pi_\mathcal{R}(a_t|s_t)}{\hat{\pi}_\beta(a_t|s_t)}\right] \\
&\geq Q^{c_i}(s_t, a_t) + \chi(I - \gamma P^\pi)^{-1}\frac{\pi_\mathcal{R}(a_t|s_t)}{\hat{\pi}_\beta(a_t|s_t)} - (I - \gamma P^\pi)^{-1}\frac{C_{r,p,\delta} C_{max}}{(1-\gamma)\sqrt{|\mathcal{D}(s_t, a_t)|}}.
\end{aligned} \tag{62}$$

We guarantee the upper bound of $\hat{Q}^{c_i}$ via choosing an appropriate value of $\chi$ based on the Eq. (62). Beside the $(I - \gamma P^\pi)^{-1}$ is a matrix of non-negative entries.

$$\chi \geq \frac{C_{r,p,\delta} C_{max}}{(1-\gamma)\sqrt{|\mathcal{D}(s_t, a_t)|}}\left[\frac{\pi_\mathcal{R}(a_t|s_t)}{\hat{\pi}_\beta(a_t|s_t)}\right]^{-1}. \tag{63}$$

## C  Experimental Details

### C.1  Dataset Details

As shown in Fig. 6, we utilize the online policy CVPO to collect sample data with different $\varphi$ values during three distinct stages of the training process. The variable $\varphi$ represents the proportion of safe trajectories in the collected samples compared to the total number of sample trajectories. Behavior 1 corresponds to $\varphi = 0.4$, behavior 2 corresponds to $\varphi = 0.6$, and so on. To ensure that the collected sample data satisfies the designated $\varphi$ value, we incorporate an additional trajectory filtering policy into the data collection procedure. Furthermore, the total number of steps for each category of samples collected is $1e7$.

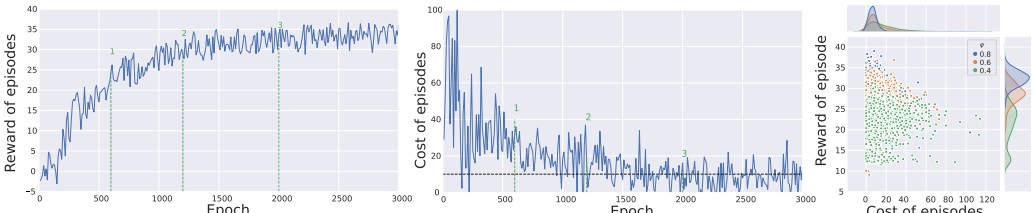

Figure 6: Collect datasets with different values $\varphi$ in the *Point-Button* task.

## C.2  Experimental Task Details

**The *Point-Goal* and *Car-Goal* task.** The objective of the agent is to reach a goal while avoiding surrounding obstacles. Once the agent reaches the correct goal, the environment randomly generates a new goal. The agent receives a reward when it approaches or reaches the goal. There are hazards and vases as obstacles in the environment, and the agent incurs a cost when encountering them. Hazards are immovable, while vases can be moved. The observation state is the relative pose between the agent and obstacles and goals, represented by a point-based pseudo-LIDAR. We select two tasks, namely *Point-Goal* and *Car-Goal* tasks [26, 29]. These tasks are performed by the *Point* and *Car* agents, respectively. The *Point* agent is a robot constrained to a 2D plane, capable of rotation as well as forward and backward movement. The *Car* agent is a slightly more complex robot with two independent parallel wheels and a freely rolling rear wheel.

Table 1: The hyper-parameters of the VOCE algorithm model. The variables $s$ and $a$ represent the dimensions of the state and action, respectively.

| Categories | Hyper-parameters | Value |
|---|---|---|
| Policy network | Sizes | $s \times 256 \times 256 \times a$ |
| | Activation function | ReLu |
| | Number of networks | 2 |
| | Learning rate | 2.0e-3 |
| | Optimizer | Adam |
| Q-value network of reward | Sizes | $(s + a) \times 256 \times 256 \times 1$ |
| | Activation function | ReLu |
| | Number of networks | 4 |
| | Learning rate | 1.0e-3 |
| | Optimizer | Adam |
| Q-value network of cost | Sizes | $(s + a) \times 256 \times 256 \times 1$ |
| | Activation function | ReLU |
| | Number of networks | 4 |
| | Learning rate | 1.0e-3 |
| | Optimizer | Adam |
| Others | Batch size | 600 |
| | Discount factor $\gamma$ | 0.99 |
| | KL threshold $\epsilon$ | 0.1 |
| | The clip of the dual variable $\eta$ | [1.0e-6, 1.0e5] |
| | The clip of the dual variable $\lambda$ | [1.0e-6, 1.0e5] |
| | The clip of the trade-off factor $\kappa$ | [0, 1.0e6] |
| | The clip of the trade-off factor $\chi$ | [0, 1.0e6] |

**The *Point-Button* and *Car-Button* task.** The objective of the agent is to navigate around stationary and moving obstacles in the environment and press one of the multiple target buttons. The task is similar to the goal task, where the agent receives a reward for either approaching or pressing the target button. The *Button* task involves the presence of dynamic obstacles that move rapidly along predefined trajectories. As the agent incurs costs upon colliding with these dynamic obstacles, the *Button* task is more challenging compared to the *Goal* task. In the *Button* task, we select two tasks: the *Point-Button* task and the *Car-Button* task [26, 29].

## C.3 Hyper-parameters

As shown in Table 1, we share the hyperparameters of the CEVO algorithm to facilitate reproducibility. Furthermore, we will share the code on GitHub at a later time.

## C.4 Datasets for the Preliminaries Section

In the preliminaries section, we employed a dataset with $\varphi = 1.0$ to evaluate offline RL methods. Additionally, we utilized a dataset with $\varphi = 0.8$ to evaluate off-policy safe RL methods. As shown in Fig. 7, $\varphi = 1.0$ indicates that the dataset includes samples with not only high rewards but also costs within the cost limit. In addition, $\varphi = 0.8$ indicates that the dataset includes samples with high rewards, while also containing samples that exceed the cost limit.

## C.5 Datasets for the Ablation Experiment

Fig 7 presents the datasets collected under the same policy but with different $\varphi$ values. To ensure the acquisition of datasets that satisfy the $\varphi$ values, we incorporate a trajectory filtering strategy into the data collection procedure and introduce random actions. Concretely, when the random value is less than the predefined threshold $\varepsilon$, we employ random actions instead of the actions provided by the policy.

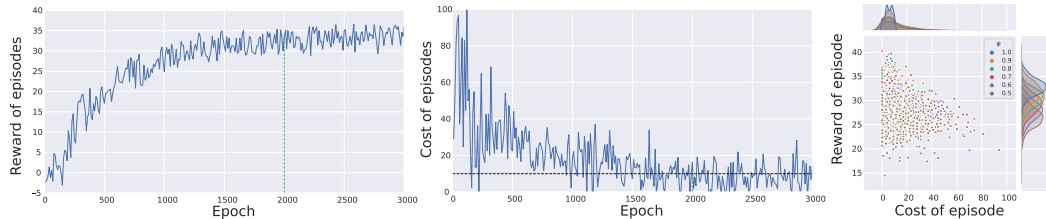

Figure 7: Collect datasets with the same policy and different values $\varphi$ in the *Point-Button* task.

## C.6 The Ablation Experiment of the Parameters

We intuitively believe that the value of the parameter $\epsilon$ in the objective 8 affects the performance of the VOCE algorithm. Therefore, we conduct ablation experiments on this parameter. The results shown in Fig 8 indicate that setting the parameter $\epsilon$ too small reduces the convergence speed of the policy and even decreases the algorithm's performance. Conversely, when the parameter $\epsilon$ is set to larger, it may lead to instability in the policy training process. Furthermore, from the illustrated results, we can conclude that within this range of parameter $\epsilon \in [0, 1]$, it ensures the policy's good convergence speed and stability. Note that the parameter $\epsilon \in [0, 1]$ represents a feasible range rather than an optimal interval.

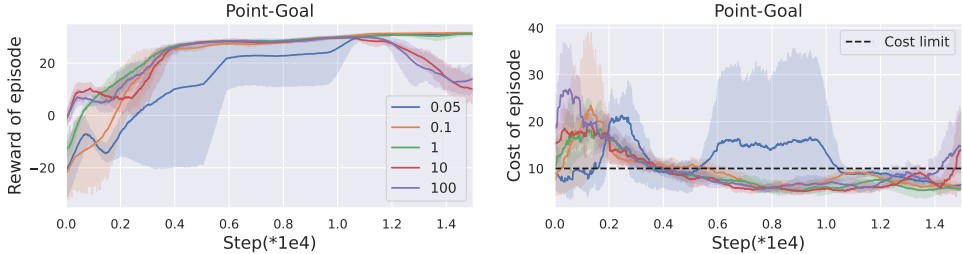

Figure 8: Reward and cost curves of VOCE in the *Point-Goal* task for different values of the parameter $\epsilon$. The parameter $\epsilon$ is the KL-divergence threshold between the variational distribution $q(a_t|s_t)$ and the parameterized policy $\pi_\theta(a_t|s_t)$.

## C.7 The trade-off factors

Regarding the trade-off factors $\kappa$ and $\chi$, we adopt a dual-tuning approach with gradient-based adaptation instead of a manual setting. To facilitate the observation of changes in the trade-off factors $\kappa$ and $\chi$, we record the variation curves of these parameters during the training process. The results of Fig. 9 indicate that as the model gradually converges, the trade-off factors $\kappa$ and $\chi$ gradually decrease, eventually approaching zero.

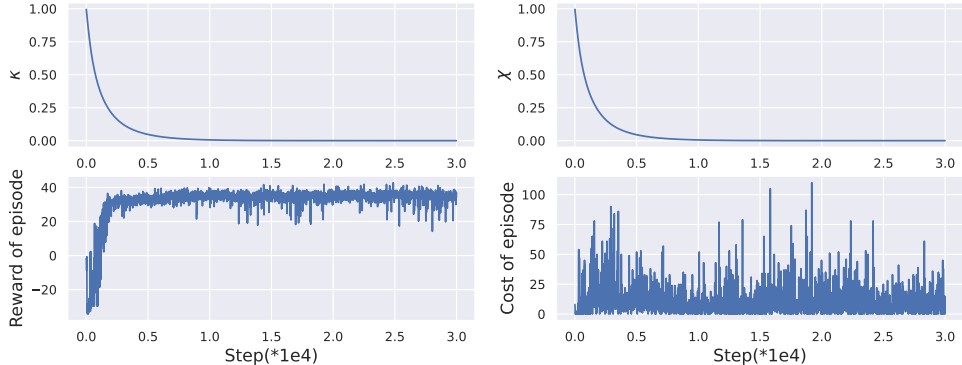

Figure 9: The trade-off factors $\kappa$ and $\chi$, are considered along with the curves of rewards and costs during the training process in task *Point-button*. Both the trade-off factors $\kappa$ and $\chi$ are adaptively adjusted using the dual-tuning approach through gradient-based.

## C.8 The time costs of training and testing

We recorded the time costs of various modules of the VOCE and the baseline algorithm during both training and testing. The results presented in Table 2 indicate that during training, the VOCE algorithm incurs additional time costs per step compared to the baseline algorithm due to the execution of additional networks and optimization parameters. However, during the testing phase, as only the policy network is executed, the time cost of VOCE is similar to the baseline algorithm. Furthermore, the execution time per step during testing is significantly lower than $1ms$, meeting the real-time requirements for robot and autonomous driving control. Additionally, the results from Table 3 reveal that the evaluation time cost of the Q-values is relatively high due to the computation of multiple target Q-values and gradients.

Table 2: The time costs per single step of the algorithm during training and testing processes. The units for the numerical values are all in seconds.

| Algorithms | Training(s) | Testing(s) |
|---|---|---|
| VOCE | 3.1498 | 3.0665 × 1e-4 |
| C-CRR | 0.0891 | 1.5140 × 1e-4 |
| COptiDICE | 0.0358 | 1.4316 × 1e-4 |
| BCQ-Lag | 0.1104 | 2.7976 × 1e-4 |

Table 3: The time costs during the training processes of various modules in VOCE. The units for the numerical values are all in seconds.

| Critic | | Actor | | |
|---|---|---|---|---|
| $Q^r(s_t, a_t)$ | $Q^c(s_t, a_t)$ | $(\lambda, \eta)$ | $q(a_t\|s_t)$ | $\pi_\theta(a_t\|s_t)$ |
| 1.2728 | 1.2668 | 0.0026 | 0.5533 | 0.0543 |

# D Limitations and Future Work

The setup of this work involves learning safety-constrained policies from offline data without interacting with the environment. Therefore, both the size and quality of the dataset directly impact

the algorithm's performance. Moreover, in the offline setting, the available sample data is limited, making it challenging to adequately represent the state transition matrix $P$, in a non-stationary environment. Consequently, the VOCE algorithm struggles to learn high-reward policies that satisfy safety constraints in non-stationary environments.

