# OpenReview forum: "VOCE: Variational Optimization with Conservative Estimation for Offline Safe Reinforcement Learning"
_NeurIPS.cc/2023/Conference — NeurIPS 2023 poster_

### Official Review · Reviewer_wQ6W · 2023-07-02

**Soundness:** 3 good
**Presentation:** 3 good
**Contribution:** 2 fair
**Rating:** 7
**Confidence:** 4

**Summary:**

This paper utilizes probabilistic inference to address the problem of offline safe RL by introducing non-parametric variational distributions.
 Pessimistic estimation of Q-values are used to avoid extrapolation errors caused by OOD actions. Extensive comparative numerical experiments are carried with respect to both reward and cost curve.

**Strengths:**

1. The paper has a solid chain of theoretical derivation.
2. The numerical results of VOCE seems promising.


**Weaknesses:**

1. The paper is not well-organized written. For example, several notations are not defined before used or at least not easy to find. For example, Q^{r}(s,a) used in equation 7, \pi_{M}(a_t|s_t) in equation 13. And it's more clear to have a complete summary of the main algorithm  VOCE, rather than deriving literally step by step.

2. Some claims in the paper need further theoretical explanation or numerical experiments to clarify.

3. It's not well studied how key tuning parameters affect the numerical conclusion.

**Questions:**


1. The algorithm involves lots of parameters to tune, like \epsilon in equation 11, \chi in equation 17. I am a little bit suspicious of how robust those parameters affect the numerical results. Maybe some ablation study is needed to further clarify.

2. Why pessimistic estimation of Q-values can help avoid extrapolation errors caused by OOD actions. I don't get the intuition here. More theoretical analysis / explanation or numerical experiments are needed.

**Limitations:**

No negative societal impact of their work is seen.

---

> ### Author Rebuttal · Authors · 2023-08-09
>
> Dear review wQ6W:
> Thank you for the insightful comments which help us improve the paper. We'll answer your questions one by one below. We are also very honored to share our understanding with you.
> + __Q1: "The paper is not well-organized written. $\cdots$ rather than deriving literally step by step." from the first weakness.__
> __A1:__ Sorry for our unclear presentation. The $Q^{r}(s_t,a_t)$ represents the reward Q-value of taking action $a_t$ in state $s_t$. The $\pi_{\mathcal{M}}(a_t|s_t)$ refers to the policy corresponding to unseen action-state pairs. We have double-checked all variables in the manuscript and provided definitions for variables such as $\rho(s_0)$, $\epsilon$. The variable $\rho(s_0)$ denotes the distribution of the initial state $s_0$. The $\epsilon$ represents the KL divergence threshold between the variational distribution $q(a_t|s_t)$ and the policy $\pi(a_t,s_t)$.  Then we will provide definitions for any undefined variables in the final version. We have presented the pseudo-code for the VOCE algorithm in Appendix A.1 of the manuscript. To enhance the clarity of the paper's structure, we will provide a summary and implementation details of the algorithm around its pseudo-code in the final version's appendix.
> + __Q2:"Some claims in the paper need further theoretical explanation or numerical experiments to clarify."__
> __A2:__ Thank you for your suggestion. If you have any further questions or concerns about our work, please do not hesitate to let us know; we would be more than happy to provide clarifications. Additionally, we have done ablation experiments concerning the manually tuned hyperparameter $\epsilon$, and the results are depicted in Fig.1. The results displayed in the figure indicate that setting the parameter $\epsilon$ too small would reduce the convergence speed of the policy and may even diminish the algorithm's overall performance. Conversely, when $\epsilon$ is set too large, policy instability can arise. The experimental outcomes depicted in the graph demonstrate that within this range of $\epsilon \in [0.1,1]$, it ensures a favorable balance between convergence and stability for the policy. The parameter $\epsilon \in [0.1,1]$ represents a feasible range rather than an optimal interval.
> + __Q3:  "It's not well studied how key tuning parameters affect the numerical conclusion."__
> __A3:__ We thank you for your pointing out this issue. Regarding the manually tuned parameter $\epsilon$, we have done ablation experiments and the results are presented in Fig.1. Additionally, Regarding the hyperparameters $\kappa$ and $\chi$ mentioned in the manuscript, setting them too small makes it difficult to guarantee the algorithm's performance theoretically, while setting them too large leads to conservative behavior and a decrease in algorithm performance. Therefore, we adopt a dual-tuning approach with gradient-based adaptation instead of a manual setting.  To facilitate the observation of the variations in trade-off factors $\kappa$ and $\chi$, we have documented the evolution curves of these factors during the training process. Furthermore, the variation curve of the trade-off factor $\kappa$ and $\chi$ during the model training process is depicted in Fig.2 . The results shown in the figure indicate that as the model gradually converges, the trade-off factor $\kappa$ and $\chi$ diminishes progressively and eventually stabilizes at a value greater than zero.
> + __Q4:"The algorithm involves lots of parameters to tune, $\cdots$ Maybe some ablation study is needed to further clarify." from the first question.__
> __A4:__ We thank you for your pointing out this issue. Regarding the manually set hyperparameter $\epsilon$, we have done ablation experiments as shown in Fig.1. The rules governing its configuration have been analyzed in Q2. Additionally, $\chi$ is adjusted using a dual-tuning approach with the gradient descent strategy, not manually set.  To facilitate the observation of the variations in trade-off factors $\kappa$ and $\chi$, we have documented the evolution curves of these factors during the training process. Furthermore, we have done a detailed analysis of its variation pattern in Q3.
> + __Q5: "Why pessimistic estimation of Q-values can help avoid extrapolation errors caused by OOD actions. I don't get the intuition here. More theoretical analysis/ explanation or numerical experiments are needed. "__
> __A5:__  Interesting and insightful question. The pessimistic conservative estimation of Q-values is rooted in a pessimistic perspective that purely assumes that OOD actions not present in the sample data possess lower rewards and higher costs. Building upon this intuitive notion, it involves minimizing or maximizing the reward Q-values and cost Q-values associated with OOD actions during the Q-value assessment process. This approach aims to induce lower estimated values for reward Q-values concerning unseen actions and higher estimated values for cost Q-values pertaining to unseen actions. Ultimately, this results in the policy reducing the probability of producing out-of-distribution (OOD) actions, thereby mitigating the extrapolation error induced by OOD actions. Furthermore, this viewpoint is substantiated by existing relevant theoretical underpinnings [1]. On the other hand, our manuscript has done ablation experiments concerning conservative estimates for $Q^{r}$ and $Q^{c_i}$. The experimental findings also indicate a notable enhancement in algorithmic reward returns due to the conservative estimate for $Q^r$, and a certain reduction in the probability of policy constraint violation attributable to the conservative estimate for $Q^{c_i}$.
>
>
> [1] Aviral Kumar, Aurick Zhou, George Tucker, and Sergey Levine. Conservative q-learning for offline reinforcement learning. Advances in Neural Information Processing Systems, 33:1179–1191, 2020.

---

> > ### Comment · Reviewer_wQ6W · 2023-08-18
> >
> > Thanks for the detailed reply.  I decide to increase my score to acceptance.

---

> > > ### Author Response · Authors · 2023-08-19
> > > **Response**
> > >
> > > Dear review wQ6W:
> > > We greatly appreciate your patient reply and extend our gratitude again for the valuable suggestions you have provided to enhance our work.

---

### Official Review · Reviewer_kEdv · 2023-07-03

**Soundness:** 3 good
**Presentation:** 3 good
**Contribution:** 3 good
**Rating:** 7
**Confidence:** 4

**Summary:**

This paper introduces an interesting algorithm for offline safe RL, called Variational Optimization with Conservative Estimation (VOCE). The primary challenge tackled by the authors is the influence of safety constraints and out-of-distribution (OOD) actions, which often hamper the optimization of high-reward policies while maintaining safety. Traditional methods, such as linear programming and exploration-evaluation methods, have shown their limitations in coping with these challenges. In response, the authors propose a probabilistic inference-based framework that leverages Expectation and Maximization (EM) to provide more flexible policy optimization. It employs pessimistic estimation methods to calculate the Q-value of cost and reward, thus mitigating extrapolation errors due to OOD actions.

**Strengths:**

1. This works is an interesting extension of off-policy safe RL (particularly CVPO) to the offline setting. The application of probabilistic inference to the problem of offline safe RL introduces non-parametric variational distributions to replace parameterized policies, improving the flexibility for optimizing safe policies.
2. The paper's methodology is clear and well-presented. The derivation of upper and lower bounds for Q-value estimation using a pessimistic estimation approach, which is then utilized to estimate Q-values of costs and rewards to mitigate extrapolation errors from OOD actions.
3. Experiments demonstrate the superiority of the VOCE algorithm over baseline methods in terms of safety.

**Weaknesses:**

1. It seems that many claims and arguments are directly derived from CVPO, such as Propositions 3.4 and 3.5. The authors should properly cite the references accordingly. In addition, some other recent offline safe RL [methods](https://arxiv.org/abs/2302.07351) and [benchmarks](https://arxiv.org/abs/2306.09303) could be discussed in the paper.
2. I am not sure whether the EM optimization objectives in CVPO could be directly applied to the offline setting, since the introduction of the KL constraints in the E-step and M-step aims to regularize the policy update such that the behavior policy will not deviate too much and cause catastrophic failure in the online setting; however, in the offline setting, since the policy will not interact with the environment, this constraint seems not very necessary? Would regularizing the KL constraints between the variational distribution and the behavior policy in the datasets (with some density estimations for the behavior policies) be a better choice?

**Questions:**

1. I think one major challenge for these constrained optimization-based approaches for offline safe learning is to accurately estimate the Q-value functions, since a subtle inaccuracy will be amplified by comparing Qc estimates and the target threshold after more and more optimization iterations. So I am curious how the authors think and tackle this problem, apart from using pessimism. In fact, pessimism alone can not solve the Q value mismatch between the current optimized policy and the behavior policy in the datasets.
2. Following up on the above question, I wonder how VOCE would perform in more diverse datasets with a wider span of trajectories over the cost-reward-return spaces. For example, collecting datasets with varying thresholds, hyper-parameters, and algorithms, as shown in [this](https://arxiv.org/abs/2306.09303) work.

**Limitations:**

I did not find related discussions.

---

> ### Author Rebuttal · Authors · 2023-08-09
>
> Dear review kEdv:
> Thank you for your all valuable suggestions and meticulous comments. We will incorporate your suggestions in the revision. Below we respond to your key concerns point by point. Please let me know if there are any further questions.
> + __Q1: "It seems that many claims and arguments are directly derived from CVPO, such as Propositions 3.4 and 3.5. The authors should properly cite the references accordingly."__
> __A1:__ Thank you for your suggestions. This work of CVPO[1] has provided profound inspiration for our work. We have cited and discussed it in Section 3.2 of our manuscript. Furthermore, we will supplement the discussion in the final version regarding the proposition that "Similar theoretical foundations can be found in the online safe RL of CVPO[1], as outlined in Propositions 3.4 and 3.5."
> + __Q2: "In addition, some other recent offline safe RL methods and benchmarks could be discussed in the paper."__
> __A2:__ Following your suggestion, we will incorporate a discussion on the CDT[2] algorithms in the related work section of the final version. Additionally, we will provide necessary comparative experiments based on the DSRL[3] dataset in the Appendix. Due to time constraints, we have done experiments of the VOCE algorithm in three scenarios of the DSRL dataset, and the experimental results are presented in Table 1 (The table is available in the supplementary PDF）. Furthermore, we consider CDT to be a highly meaningful work that leverages the return-conditioned sequence modeling framework to facilitate zero-shot adaptation to various constraint thresholds during deployment while ensuring both safety and high rewards.
> + __Q3: "I am not sure whether $\cdots$ this constraint seems not very necessary? " from the second weakness.__
> __A3:__ Insightful question. Introducing the KL divergence constraint during the optimization of the optimal variational distribution $q(a_t|s_t)$ can mitigate policy oscillations caused by the iteration of Lagrange multipliers in the optimization process. Furthermore, as we have already introduced the KL divergence constraint between the optimal variational distribution $q(a_t|s_t)$ and the previous iteration's policy $\pi_{\theta}(a_t|s_t)$ during the process of solving for the optimal variational distribution $q(a_t|s_t)$, imposing an additional constraint on the distance between the latest policy and the previous policy could lead to slow policy convergence and even performance degradation. Hence, we have not incorporated the KL divergence constraint in the process of updating the parameterized policy.
> + __Q4："Would regularizing the KL constraints between the variational distribution and the behavior policy in the datasets (with some density estimations for the behavior policies) be a better choice?"__
> __A4:__ Interesting and insightful question. You have provided a meaningful approach that utilizes KL divergence to constrain the distance between the variational distribution and the behavioral policy. This method aims to reduce the discrepancy between the learned policy and the behavioral policy based on sample data, potentially improving the algorithm's performance on expert and safe datasets. However, for non-expert and non-safe datasets, this approach might hinder the variational distribution from deviating away from unsafe behaviors, making it challenging for the final policy to satisfy safety constraints unless we specifically train our policy using expert safety data.
> + __Q5: "I think one major challenge for $\cdots$ optimized policy and the behavior policy in the datasets."  from the first question.__
> __A5:__  I fully agree with your perspective. One of the main challenges in offline safe reinforcement learning based on off-policy methods is ensuring the accuracy of Q-value estimation. Indeed, relying on strict upper bound estimation for $Q^{c_i}$ can lead to error accumulation. To mitigate the issue of Q-value mismatch between the current policy and the behavior policy after multiple iterations, we place full trust in the action-state pairs present in the sample data. Therefore, during the Q-value of cost $Q^{c_i}$ update, we maximize the $Q^{c_i}$ of action-state pairs present in the sample data and incorporate this trick into the VOCE code.
> + __Q6: "Following up on the above question, I wonder how VOCE would perform in more diverse datasets with a wider span of trajectories over the cost-reward-return spaces. For example, collecting datasets with varying thresholds, hyper-parameters, and algorithms, as shown in this work."__
> __A6:__ Thank you for your suggestion. You mentioned that the benchmark provides an excellent testing methodology, which evaluates the algorithm's performance by testing it under various cost thresholds. This method effectively captures the algorithm's performance across different cost thresholds and hyperparameters. Therefore, we are considering including our algorithm's performance on this benchmark in the final version. Currently, we have included the results from several tasks we have conducted, as shown in Table 1 (The table is available in the supplementary PDF）.  Additionally, another recent work of ours also considers evaluating our algorithm's performance based on this benchmark.
>
>
> [1] Zuxin Liu, Zhepeng Cen, Vladislav Isenbaev, Wei Liu, Steven Wu, Bo Li, and Ding Zhao. Constrained variational policy optimization for safe reinforcement learning. In International Conference on Machine Learning, pages 13644–13668. PMLR, 2022.
> [2] Zuxin Liu, Zijian Guo, Yihang Yao, Zhepeng Cen, Wenhao Yu, Tingnan Zhang, and Ding Zhao. Constrained decision transformer for offline safe reinforcement learning. arXiv preprint arXiv:2302.07351, 2023a.
> [3] Zuxin Liu, Zijian Guo, Haohong Lin, Yihang Yao, Jiacheng Zhu, Zhepeng Cen, Hanjiang Hu, Wenhao  Yu, Tingnan Zhang, Jie Tan, et al. Datasets and benchmarks for offline safe reinforcement learning.  arXiv preprint arXiv:2306.09303, 2023b.

---

> > ### Comment · Reviewer_kEdv · 2023-08-16
> > **Response**
> >
> > Thanks for the detailed reply and clarification. Most of my concerns are well-addressed, and I am hoping to see the revision with better paper presentations. I decide to increase my score to acceptance.

---

> > > ### Author Response · Authors · 2023-08-16
> > > **Response**
> > >
> > > Dear review kEdv:
> > > We greatly appreciate your patience reply. We are also grateful for your recognition of our efforts and the positive evaluation of our work. Once again, we extend our gratitude for your valuable feedback. Your suggestions are of great value in enhancing the quality of our work.

---

### Official Review · Reviewer_GTEu · 2023-07-07

**Soundness:** 3 good
**Presentation:** 2 fair
**Contribution:** 2 fair
**Rating:** 5
**Confidence:** 3

**Summary:**

This paper applies CQL to offline safe RL tasks. It introduces a non-parametric policy to search for the actions that satisfy the constraints. Lagrangian multipliers are introduced for constraining the KL divergence w.r.t. the current policy and the additional constraints. CQL-style additional terms are introduced to avoid extrapolation issues during the offline training. The method is evaluated at four safe RL environments and demonstrates that the proposed approach can achieve better rewards while better approaching the cost limits.

**Strengths:**

- The paper proposed a solid approach that consists necessary ingredients for offline-safe RL. The conservative estimation follows CQL and should be able to mitigate the issues in offline RL.
- An optimal variational distribution is introduced to improve the policy in a non-parameterized way. This step is novel in offline safe RL.
- The paper proposed an offline safe RL benchmark and made a careful study.

**Weaknesses:**

- The novelty of the method could be more extensive. The method mostly combines existing methods. The variational method, non-parameterized policy estimation, constraint formulation, and conservative Q estimation are all standard methods.
- Some mathematical notations are not clear. For example, Eq (19) and Eq (20) are quite misleading. I think it should be log \pi_\theta(a_t|s_t) in Eq(2).
- If my understanding is correct, the proposed method is not able to reach the cost limitation in several point-button and car-button environments, which suggests the inadequate of the proposed method.

**Questions:**

I would like to know the reason that the authors chose to follow CQL instead of other offline RL algorithms. Since there is already a KL term in the optimization objective, I think Behavior cloning or advantage-weighted approach, which directly constrains the KL divergence between the learned policy and the behavior policy, is more straightforward here and may achieve higher performance in practice. Is there any reason to consider the conservative Q estimation rather than other choices?

**Limitations:**

not discussed

---

> ### Author Rebuttal · Authors · 2023-08-09
>
> Dear review GTEu:
> Thank you for your suggestions and constructive comments. We will incorporate your suggestions in the revision. Below we respond to your key concerns point by point. Please let me know if there are any further questions.
> + __Q1:" The novelty of the method could be more extensive. The method mostly combines existing methods. The variational method, non-parameterized policy estimation, constraint formulation, and conservative Q estimation are all standard methods."__
> __A1:__ We agree that our VOCE method has been inspired by these advanced RL techniques. Although our method is developed from the existing techniques,  we extend it to the setting of offline safe reinforcement learning, thereby significantly broadening their applicability. Furthermore, the proposed method achieves superior performance in offline safe reinforcement learning tasks compared to the current state-of-the-art approaches. Now, we proceed to delineate the primary contributions of our VOCE method and highlight its key distinctions from existing approaches in the following aspects.
> (1) We propose a novel variational optimization with conservative estimation for offline safe reinforcement learning based on variational inference and pessimistic estimation methods to solve the problem of optimizing safety policies in the offline dataset. To the best of my knowledge, this is the first time that variational inference has been introduced to solve the task of offline safe reinforcement learning.
> (2) Subsequently, we extend the pessimistic estimation to the offline safe reinforcement learning setting, deriving for the first time an upper bound for Q-value. We employ this upper bound to estimate the cost Q-value $Q^{c_i}$, thereby avoiding underestimation of the cost Q-value $Q^{c_i}$.
> (3) In the field of offline safe reinforcement learning, we introduce for the first time the validation of algorithm performance using sample data under various safety ratios $\varphi$. Furthermore, extensive experiments demonstrate that our algorithm significantly offers competitive performance, particularly in terms of safety, as evidenced by the results.
> + __Q2:" Some mathematical notations are not clear.  For example, Eq (19) and Eq (20) are quite misleading.  I think it should be $log{\pi}_{\theta}(a_t|s_t)$  in Eq(20)."__
> __A2:__ We are sorry for the unclear mathematical representation, and we greatly appreciate your suggestions.  We have revised Eq.(19) and (20) according to your suggestions, as shown below:
> $\mathcal{L}(\theta)=\max \mathbb{E}\_{\tau\sim q} \left[-\alpha D_{KL}(q(\cdot|s_t)||\pi_{\theta}(\cdot|s_t))\right]= \max \alpha \mathbb{E}\_{\rho(s_0)} \mathbb{E}\_{q(a_t | s_t)} [\log {\pi_{\theta}(a_t|s_t)}-\log q(a_t|s_t)]$
> $\mathcal{L}(\theta)=\max \mathbb{E}\_{\rho(s_0)}\mathbb{E}\_{q(a_t | s_t)}  [\log {\pi_{\theta}(a_t |s_t)}]$
> Additionally, we have double-checked all the equations in the manuscript and will update any inaccurate representations in the final version.
> + __Q3:"If my understanding is correct, the proposed method is not able to reach the cost limitation in several point-button and car-button environments, which suggests the inadequate of the proposed method."__
> __A3:__ Yes, your understanding is correct.  In the *Point-button* and *Car-button* scenarios, there are continuously moving obstacles, which significantly increase the dimensionality of the dynamic transition matrix $P$ compared to the *Point-goal* and *Car-goal* scenarios. However, the size of our sample dataset in all scenarios is 1e7 action-state pairs, which leads to a significant increase in OOD actions in the *Point-button* and *Car-button scenarios*, thereby affecting the algorithm's performance. Furthermore, even though the cumulative costs in these two scenarios do not meet the safety constraints, the cumulative costs achieved by the VOCE algorithm are lower than those of the previous state-of-the-art algorithms. This suggests that the performance of the VOCE algorithm surpasses that of the previous state-of-the-art algorithms. To ensure the policy's adherence to safety constraints in both the *Point-button* and *Car-button* scenarios, we can explore the following approaches:(1) Increasing the amount of sample data in these scenarios to mitigate the impact of OOD actions during policy training. (2) Increasing the minimum pruning threshold for hyperparameter $\chi$, however, may lead to a significant reduction in reward values.
> + __Q4: "I would like to know the reason that the authors chose to follow CQL instead of other offline RL algorithms. Since there is already a KL term in the optimization objective, I think Behavior cloning or advantage-weighted approach, which directly constrains the KL divergence between the learned policy and the behavior policy, is more straightforward here and may achieve higher performance in practice. Is there any reason to consider the conservative Q estimation rather than other choices? "__
> __A4:__   Insightful question. You have presented an excellent approach by constraining the target policy to the behavior policy using the KL divergence. Under the safety of expert sample data, this method enables the target policy to achieve high rewards while adhering to safety constraints. However, it is challenging to guarantee that the policy satisfies safety constraints, especially with non-expert and unsafe datasets such as $\varphi =\{0.4,0.6 \} $.  We estimate $Q^{r}$ and $Q^{c_i}$ respectively through the upper and lower bounds of Q-value estimation, which can theoretically ensure that the strategy meets the safe constraints, and also avoid the need to add additional constraints to increase the instability of model training when solving the optimal variational distribution.

---

### Official Review · Reviewer_cmps · 2023-07-07

**Soundness:** 3 good
**Presentation:** 4 excellent
**Contribution:** 3 good
**Rating:** 7
**Confidence:** 3

**Summary:**

The paper presents a variational approach to offline safe reinforcement learning (RL), where the class of variational distributions is the set of policies which satisfy the cost constraints. It is shown how to obtain the closed-form solution for the optimal variational distribution, and how to extract a parametric policy given the variational distribution. To guarantee that the reward Q function is not overestimated and the cost Q function is not underestimated, the tools of Conservative Q Learning (CQL) are incorporated.

**Strengths:**

* The variational formulation is novel to my knowledge and may inspire additional work in this area.
* VOCE performance is good compared to existing algorithms, both in terms of high reward and satisfying the cost constraint.
* The paper is generally clear and not hard to read.

**Weaknesses:**

* There is no discussion of the runtime of the algorithm, which I imagine is comparatively high, given that you have to solve an optimization problem (Eqn. 11) at every step.
* While the final hyperparameters are provided, we are not told how they were tuned, nor do we have a sense of how sensitive VOCE’s performance is to their values.

**Questions:**

* How are the trade-off factors ($\kappa$, $\chi$) determined? Are you using a dual tuning method akin to the $\alpha$ in CQL?
* In Eqn. (20), should be $a_t$ instead of $\cdot$?

**Limitations:**

No, limitations are not addressed.

---

> ### Author Rebuttal · Authors · 2023-08-09
>
> Dear review cmps:
> Thank you for your all valuable suggestions and meticulous comments. We will incorporate your suggestions in the revision. Below we respond to your key concerns point by point. Please let me know if there are any further questions.
>
> + __Q1: "There is no discussion of the runtime of the algorithm, which I imagine is comparatively high, given that you have to solve an optimization problem (Eqn. 11) at every step."__
> __A1:__ We greatly appreciate your raising this question. We have documented the single-step time consumption of the VOCE and existing methods during both training and testing processes. Additionally, we have recorded the time consumption of each module of the VOCE algorithm during the training process. The recorded results are presented in Tables 1 and 2, respectively. The results shown in Table 1 indicate that during the training process, the inclusion of additional networks and optimization parameters in VOCE indeed leads to a noticeable increase in the time consumption per step compared to other methods. However, during the testing process, since only the policy network needs to be executed, the per-step time consumption of the VOCE algorithm is only slightly different from other algorithms. Furthermore, the execution time per step during testing is significantly below 1 ms, which satisfies the real-time requirements of practical applications such as robot control and autonomous driving. Furthermore, as evident from the results in Table 2, higher time consumption is observed during Q-value evaluation due to the computation of multiple target Q-values and gradients. On the other hand, the time consumption is relatively lower for the Lagrange multipliers $\lambda, \eta$ as they are updated only once per step utilizing the gradients.
> __Table 1 The computational time per single step of the algorithm during training and testing processes.__
> |Algorithms|Training(s)|Testing(s) |
> | ---------| --------- | --------- |
> |VOCE      |3.1498     |3.0665×1e-4|
> |C-CRR     |0.0891     |1.5140×1e-4|
> |COptiDICE |0.0358     |1.4316×1e-4|
> |BCQ-Lag   |0.1104     |2.7976×1e-4|
>
> __Table 2 Time consumption during the training processes of various modules in VOCE.__
> | $Q^{r}(s_t,a_t)$ | $Q^{c}(s_t,a_t)$  |  $(\lambda, \eta)$  | $q(a_t\|s_t)$ | $\pi_{\theta}(a_t\|s_t)$      |
> |:---------:|:---------:|:---------:|:---------:|:---------:|
> | 1.2728 | 1.2668 | 0.0026 | 0.5533 | 0.0543 |
> + __Q2: "While the final hyperparameters are provided, we are not told how they were tuned, nor do we have a sense of how sensitive VOCE’s performance is to their values. How are the trade-off factors ($\kappa$, $\chi$) determined? Are you using a dual tuning method akin to the $\alpha$ in CQL? "__
> __A2:__ Interesting and insightful question. Yes, we have employed a dual-tuning approach similar to CQL. We take into consideration that setting the trade-off factor $\kappa$ too large might result in a significant underestimation of the Q-values $Q^{r}$, while putting it too small could struggle to suppress the overestimation of Q-values $Q^{r}$ caused by OOD actions. Manually adjusting these parameters is time-consuming; hence, we follow the dual tuning methodology of CQL, utilizing gradient descent for adaptive parameter $\kappa$ adjustment. The tradeoff factor $\chi$ is also set using a similar method.  To facilitate the observation of the variations in trade-off factors $\kappa$ and $\chi$, we have documented the evolution curves of these factors during the training process. The variation curve of the trade-off factor $\kappa$ and $\chi$ during the model training process is depicted in Fig.2 . The results shown in the figure indicate that as the model gradually converges, the trade-off factor $\kappa$ and $\chi$ diminishes progressively and eventually stabilizes at a value greater than zero.
>
> + __Q3:"In Eqn. (20), should be $a_t$ instead of $\cdot$ ?"__
> __A3:__ Thank you for your suggestion. We have revised Eq. (20) to the following expression:
>    $\mathcal{L}(\theta)=\max \mathbb{E}\_{\rho(s_0)}\mathbb{E}\_{q(a_t | s_t)}  [\log {\pi_{\theta}(a_t |s_t)}]$
> Additionally, we have double-checked all the equations in the manuscript, and we have revised Eq.(19) to:
>   $\mathcal{L}(\theta)=\max \mathbb{E}\_{\tau\sim q} \left[-\alpha D_{KL}(q(\cdot|s_t)||\pi_{\theta}(\cdot|s_t))\right]= \max \alpha \mathbb{E}\_{\rho(s_0)} \mathbb{E}\_{q(a_t | s_t)} [\log {\pi_{\theta}(a_t|s_t)}-\log q(a_t|s_t)]$

---

> > ### Comment · Reviewer_cmps · 2023-08-21
> >
> > Thank you for your detailed response. I think the discussion of runtime would be useful context to include, even if only in the appendix.
> >
> > Regarding hyperparameters:
> > * "The results shown in the figure indicate that as the model gradually converges, the trade-off factor $\kappa$ and $\chi$ diminishes progressively and eventually stabilizes at a value greater than zero." It is not clear from the plots that $\kappa$ and $\chi$ converge to positive values – it looks like they go to zero. Perhaps it would be more apparent if you used a log-y scale for these plots?
> > * A description of how the dual tuning is performed would be a useful addition. In particular, you need to introduce additional constraints, so (i) what are these constraints, and (ii) how do you choose the hyperparameters involved in these constraints?

---

> > > ### Author Response · Authors · 2023-08-21
> > > **Response**
> > >
> > > Dear review cmps:
> > > Thank you very much for your reply. Your valuable suggestions hold significant implications for enhancing the quality of our work. We will include a discussion on the model's runtime in the appendix of the final version.
> > > __Q4: "The results shown in the figure indicate that as the model gradually converges, the trade-off factor  and  diminishes progressively and eventually stabilizes at a value greater than zero." It is not clear from the plots that  and  converge to positive values – it looks like they go to zero. Perhaps it would be more apparent if you used a log-y scale for these plots? "__
> > > __A4:__ Thank you for your suggestion. Adopting a logarithmic scale does indeed provide a clearer representation of the variation trends and stability of the trade-off factor. We have transformed the trade-off factor's variation curve into a logarithmic coordinate system. However, we are currently unable to submit images. We present the stable values of the trade-off factor in the table below.
> > > __Table 1 Real and logarithmic values of the balancing factors $\kappa$ and $\chi$__
> > > |           | 10           | 20           | 30           |   | 29960        | 29970        | 29980        | 29990        | 30000         |
> > > |-----------|--------------|--------------|--------------|---|--------------|--------------|--------------|--------------|---------------|
> > > | $\kappa$     | 0.99631976   | 0.9858102    | 0.97578514   | … | 1.55E-05     | 1.55E-05     | 1.54E-05     | 1.54E-05     | 1.53E-05      |
> > > | $\log\kappa$ | -0.003687029 | -0.014291438 | -0.02451286  | … | -11.07079701 | -11.0749356  | -11.07907    | -11.0832096  | -11.08735392  |
> > > | $\chi$       | 0.99612333   | 0.98561966   | 0.97548907   | … | 1.41E-05     | 1.40E-05     | 1.40E-05     | 1.39E-05     | 1.38E-05      |
> > > |$\log\chi$ | -0.003884204 | -0.014484739 | -0.024816323 | … | -11.17112616 | -11.17524275 | -11.17934963 | -11.18346784 | -11.18759897  |
> > >
> > > __Q5:A description of how the dual tuning is performed would be a useful addition. In particular, you need to introduce additional constraints, so (i) what are these constraints, and (ii) how do you choose the hyperparameters involved in these constraints?__
> > > __A5:__ Thank you very much for your suggestion. Dual tuning involves adapting the balancing trade-off factors through a gradient descent strategy.
> > > Concretely, building upon the optimization objective of the Bellman iteration, we introduced a trade-off factor $\kappa$. This factor facilitated the minimization of the reward Q-values of action-state pairs under the marginal distribution of unseen actions $\mathbb{E}\_{s_t\sim {D},a_t \sim \pi_{\mathcal{M}}(a_t|s_t)} Q^{r}(s_t,a_t)$, as well as the maximization of the reward Q-values of action-state pairs occurring within the sample space $\mathbb{E}\_{s_t\sim {D},a_t \sim \hat\pi_{\beta}(a_t|s_t)} Q^{r}(s_t,a_t)$. Furthermore, by taking the partial derivative of the trade-off factor $\kappa$ with respect to the Eq.(13) in the manuscript, we subsequently employed the gradient descent strategy to adaptively adjust the balancing factor $\kappa$. On the other hand, building upon the optimization objective of the Bellman iteration, we introduced the trade-off factor $\chi$ to maximize the cost Q-values of action-state pairs under the marginal distribution of unseen actions $\mathbb{E}\_{s_t\sim {D},a_t \sim \pi_{\mathcal{R}}(a_t|s_t)} Q^{c_i}(s_t,a_t)$. Furthermore, by taking the partial derivative of the trade-off  $\chi$ with respect to Eq. (17) in the manuscript, we subsequently employed the gradient descent strategy to adaptively adjust the trade-off factor $\chi$.

---

### Author Rebuttal · Authors · 2023-08-10

Dear reviewers:
We thank all reviewers for your time and suggestions, and we expect to have a further discussion. We have responded to your questions in detail accordingly. If you have further questions or concerns, we still reply before the end of the author-reviewer discussion. Thank you very much for your review time and efforts.

+ __How should the hyperparameters $\kappa$, $\chi$, and $\epsilon$ be set?__
__A:__  **(1)** Regarding the hyperparameters $\kappa$ and $\chi$ mentioned in the manuscript, setting them too small makes it difficult to guarantee the algorithm's performance theoretically, while setting them too large leads to conservative behavior and a decrease in algorithm performance. Therefore, we adopt a dual-tuning approach with gradient-based adaptation instead of a manual setting. To facilitate the observation of the variations in trade-off factors $\kappa$ and $\chi$, we have documented the evolution curves of these factors during the training process. The variation curve of the trade-off factor $\kappa$ and $\chi$ during the model training process is depicted in Fig.2 . The results shown in the figure indicate that as the model gradually converges, the trade-off factor $\kappa$ and $\chi$ diminishes progressively and eventually stabilizes at a value greater than zero.
**(2)** Regarding the manually set hyperparameter $\epsilon$, we have conducted ablation experiments as shown in Fig.1 . The results displayed in the figure indicate that setting the parameter $\epsilon$ too small would reduce the convergence speed of the policy and may even diminish the algorithm's overall performance. Conversely, when $\epsilon$ is set too large, policy instability can arise. The experimental outcomes depicted in the graph demonstrate that within this range of $\epsilon \in [0.1,1]$, it ensures a favorable balance between convergence and stability for the policy. The parameter $\epsilon \in [0.1,1]$ represents a feasible range rather than an optimal interval.

+ __Limitations and Future Work__
__A:__ The setup of this study involves learning policies that satisfy safety constraints from offline data without interacting with the environment. Consequently, the size and quality of the dataset have a direct impact on the algorithm's performance. Furthermore, within the context of the offline setting, the limited sample data available makes it challenging to adequately represent the state transition matrix p in non-stationary environments. As a result, the VOCE algorithm faces difficulties in learning high-reward policies that satisfy safety constraints in non-stationary environments. We will supplement the discussion of the limitations of this work in the final version.

---

### Decision · Program_Chairs · 2023-09-21

**Decision:**

Accept (poster)

**Comment:**

The paper tackles Safe offline RL using Conservative Q Learning in a variational optimization framework, over set of policies that satisfy the cost constraints. The reviewers are mostly positive and the biggest criticisms are around hyperparameter choice, runtime and clarity on limitations. My own assessment on the paper is that the method is a combination of several known ML ideas albeit the application to Safe offline setting is new.

There was rich discussion between the authors and the reviewers, where a lot of technical aspects of the work were clarified. My recommendation is accept based on the strength of the reviews as well as the insightful discussion.